# Phonon-driven wavefunction localization enhances room-temperature single-photon purity in large hybrid lead halide perovskite quantum dots

Leon G. Feld [1,2], Simon C. Boehme [1,2], Sebastian Sabisch[1,2], Nadav Frenkel[3], Nuri Yazdani [4], Viktoriia Morad[1,2], Chenglian Zhu[1,2], Taehee Kim [1,2], Stefano Canossa[1], Mariia Svyrydenko[1,2], Rui Tao[1,2], Maryna I. Bodnarchuk [1,2], Gur Lubin [3], Miri Kazes [3], Vanessa Wood[4], Dan Oron [3] ✉, Gabriele Rainò [1,2] ✉ & Maksym V. Kovalenko [1,2] ✉

In lead halide perovskites (APbX$_3$), the effect of the A-site cation on optical and electronic properties has initially been thought to be marginal. Yet, evidence of beneficial effects on solar-cell performance and light emission is accumulating. Here, we report that the A-site cation in soft APbBr$_3$ colloidal quantum dots (QDs) controls the phonon-induced localization of the exciton wavefunction. Insights from ab-initio molecular-dynamics simulations and single-particle fluorescence spectroscopy demonstrate that anharmonic crystal vibrations and the resulting disorder act as an additional confinement potential. Avoiding the trade-off between single-photon purity and optical stability faced by downsizing conventional QDs into the strong confinement regime, dynamical phonon-induced confinement in large organic-inorganic perovskite QDs enables bright (10$^6$ photons/s), stable ( >1 h), and pure ( > 95%) single-photon emission tunable across a wide spectral range (495-745 nm). Strong electron-phonon interaction in soft perovskite QDs provides an unconventional route toward developing scalable room-temperature quantum-light sources.

Reliable and scalable single-photon sources (SPSs) are essential for the broad adoption of quantum optical methods in computing[1], communication[2], and imaging[3]. So far, epitaxially-grown quantum dots (QDs) operated at liquid-helium temperatures excel with on-demand photon generation at exceptional single-photon purity, fluorescence rates, coherence times and indistinguishability[4]. Recently, coherent[5], indistinguishable[6] and fast single-photon emission[7] was also demonstrated for colloidal cesium lead halide perovskite QDs at similar

temperatures. Yet, capitalizing on the excellent performance of such sources is constrained by the need for cryogenic operation that limits their scalability and accessibility. Room-temperature SPSs alleviate the need for cryogenic cooling thus reducing significantly the system complexity. Amongst other room-temperature SPSs like doped organic crystals[8] or defect-based emitters[4], colloidal QDs exhibit scalable and low-cost syntheses based on simple solution chemistry, precise control over their optical properties, and ease of solution

[1]Laboratory of Inorganic Chemistry, Department of Chemistry and Applied Biosciences, ETH Zürich, Zürich, Switzerland. [2]Laboratory for Thin Films and Photovoltaics, Empa–Swiss Federal Laboratories for Materials Science and Technology, Dübendorf, Switzerland. [3]Department of Molecular Chemistry and Materials Science, Weizmann Institute of Science, Rehovot, Israel. [4]Department of Information Technology and Electrical Engineering, ETH Zürich, Zürich, Switzerland. ✉e-mail: dan.oron@weizmann.ac.il; rainog@ethz.ch; mvkovalenko@ethz.ch

**Fig. 1 | Single-photon emission from colloidal quantum dots (QDs) through conventional and disorder-induced quantum confinement. a** Illustration of the emitted photon statistics from individual colloidal QDs and their size-dependence. Grey dots indicate the photons emitted from particles with different particle sizes relative to their Bohr diameter ($d_B$). Photon statistics are evaluated via the second-order autocorrelation of the photon arrival times ($g^2(\tau)$). Single-photon purity is anticorrelated with $g^2(\tau)$ at a time delay ($\tau$) of zero. **b** Schematic drawing of disorder through random atomic displacement (blue) from the equilibrium positions (grey). **c** Illustration of weak disorder-induced wavefunction localization in a QD. The potential energy surface $U(x)$ of a QD without disorder (black dashed line) results in a probability-density distribution $\Psi_0(x)$ delocalized across the entire QD (grey area). A disordered potential (blue line) induces localization of the probability-density distribution (blue area).

processing. Comprehensive structural and compositional engineering elevated the photoluminescence (PL) characteristics of colloidal QDs to near-unity quantum yields (QY), suppressed PL fluctuations, and narrow-band emission[9–11]. Already implemented in various classical optoelectronic technologies[12–14], some of which are widely commercialized, colloidal QDs are becoming promising candidates for single-emitter applications[15–17]. However, colloidal QDs are yet to match the performance of cryogenically-cooled epitaxial QDs in terms of single-photon purity and PL stability over time.

For colloidal QDs, improvements of their single-photon purity can be obtained by reducing their physical size[18–20]. Such size confinement yields discretized energy levels, strong (multi-)exciton interaction and accelerated non-radiative Auger-Meitner[21] recombination of multiexcitons[22,23]. The resulting single-photon purity (exceeding 90% in colloidal QDs)[17,18,24] is inferred from the amplitude of the dip in $g^2(\tau)$, the second-order photon-photon correlation function, at zero delay time (Fig. 1a). However, enhancing the single-photon purity via QD downsizing limits the spectral tunability and comes at a high cost: Firstly, vulnerability to surface defects and matrix effects that cause stochastic intensity fluctuations (blinking) and photobleaching is amplified in small QDs[9,10,25]. Secondly,

increased coupling to surface vibrations broadens the emission spectrum of small QDs[9,26]. Finally, the volume scaling of the absorption cross sections translates into the need for high excitation densities, compromising the potential for on-chip optical pumping by LEDs and prompting uncorrelated background photons that contaminate single-photon emission. In conclusion, finding alternative ways toward high single-photon purities without the need for detrimental size confinement would improve the quality, stability, and versatility of colloidal QDs as room-temperature SPSs.

In search for alternatives to size confinement, we identified the coupling between electronic and vibrational degrees of freedom (vibronic coupling)[27] as an underutilized route towards confining charge carriers in semiconductor sources. Vibronic coupling broadens the emission lines of semiconductors employed in display and lighting applications, limits the overall charge mobility in crystalline and amorphous materials used in photovoltaics, and causes the loss of exciton optical coherence, a key resource for photon-based quantum computing. Nevertheless, while generally considered a detrimental interaction, vibronic coupling can mediate fascinating phenomena, such as Bose-Einstein condensation of exciton-polaritons[28], superconductivity[29], and optical cooling[30].

In crystalline materials at finite temperatures, crystal vibrations (phonons) dynamically displace atoms from their average lattice site (Fig. 1b), hereby inciting dynamic structural disorder that translates into electronic disorder through vibronic coupling[31,32]. Such dynamic disorder adds to any potentially present static disorder such as lattice site disorder in defected crystalline solids[33] or structural disorder in amorphous solids[34]. The associated disordered deformation potential may serve as an effective confinement potential that causes contracted and localized electronic wavefunctions, in some aspects resembling Anderson localization[32,35–38]. We hypothesize that the combination of phonon-induced disorder and exciton-phonon coupling can act in addition to the quantum size effect, thereby enhancing quantum confinement of (multi-)excitons and consequently single-photon purity (Fig. 1c). This approach does not come with the risk of trap state formation which usually reduces PL quantum yield and the overall performance of single-photon sources, but requires the use of materials that combine high amplitude, anharmonic vibrations at room temperature with strong exciton-phonon coupling[32].

A suitable platform to test our hypothesis is provided by lead halide perovskite (LHP) materials ($APbX_3$; A=formamidinium, methylammonium, Cs; X=Cl, Br, I) that emerged as promising solar-cell materials in part due to their high defect tolerance and facile solution processing. Colloidal perovskite QDs share this defect-tolerance and therefore achieve near-unity PLQY and relatively weak blinking without the need for delicate core-shell engineering[39,40]. Perovskite QDs have demonstrated single-photon emission at room temperature[41,42] as well as bright, coherent, and indistinguishable single-photon emission at cryogenic temperatures[5,6]. Initially, the A-site cation has largely been deemed a bystander in defining the photo-physics of LHPs due to its lack of direct electronic contribution to the band-edge states[43]. However, A-site cations can influence the structure of the $PbX_3^-$ framework and associated crystal dynamics, which, via a strong coupling of the exciton to anharmonic vibrations, also impact the optoelectronic performance[44,45]. Consequently, the A-site cation's importance for optical and electronic characteristics in LHPs is being recognized and exploited to improve, for example, LHP solar cell performance[43,46–51].

In this work, we suggest that the large dynamic structural disorder in formamidinium lead bromide ($FAPbBr_3$) QDs increases the effective quantum confinement and thereby renders even large QDs high-quality quantum emitters of single photons. Identifying disorder as an alternative route towards quantum confinement alleviates well-known size-dependent QD performance trade-offs. It extends the single-photon emission capability to QDs of sizes larger than the Bohr diameter, hereby straightforwardly improving both the source brightness and its stability. Ab-initio molecular-dynamics (AIMD) simulations pinpoint A-site-cation-controlled anharmonic vibrational modes as the origin of the structural disorder responsible for dynamic wavefunction localization and enhanced quantum confinement. The resulting enhanced Auger-Meitner recombination rate of multi-excitons even in large $FAPbX_3$ QDs enables pure single-photon emission, which is also bright, stable, and finely spectrally tuneable across the visible spectrum.

## Results and discussion
### Influence of the A-site cation on structure and optical properties
We selected $CsPbBr_3$ and $FAPbBr_3$ QDs as candidates to study the effect of the A-site cation on structural disorder because their room-temperature crystal structures and dynamics are tuned by the choice of the A-site cation. The room-temperature crystal structure of $CsPbBr_3$ is orthorhombic as determined by single-crystal X-ray diffraction (Fig. 2a). On the other hand, $FAPbBr_3$ adopts an average cubic crystal structure (Fig. 2b). Figure 2c and d display magnified views of the Pb-Br-Pb bonding of the corner-shared $PbBr_6$ octahedra. $CsPbBr_3$ and $FAPbBr_3$ differ in the Pb-Br-Pb angles as well as displacement ellipsoids that describe the static and dynamic disorder

occurring at finite temperatures. In $CsPbBr_3$, the average Pb-Br-Pb angle deviates from 180° as octahedral tilting compensates for the undersized Cs cation. Contrarily, a larger cation size and thus better size fitness of FA in the octahedral void stabilizes an average cubic structure, as evidenced by a Pb-Br-Pb angle of 180°. The shape of the displacement ellipsoids of Br reveals that in both systems octahedral tilting dominates disorder in the $PbBr_3^-$ framework, albeit the cubic $FAPbBr_3$ structure additionally exhibits larger displacement along the Pb-Br bond, 0.1910(16) Å compared to 0.157(3) Å in $CsPbBr_3$.

Single-particle PL spectroscopy can optically probe the small changes in structure and dynamics of the $PbBr_3^-$ framework that are introduced by the different A-site cations. Figure 2e displays representative PL spectra of $CsPbBr_3$ QDs and $FAPbBr_3$ QDs with edge lengths of 9.9(1.2) nm as determined by transmission electron microscopy (Supplementary Fig. 1). With edge lengths of 1.61(19) and 1.27(16) times their Bohr diameter ($d_B$; 6.16 nm for $CsPbBr_3$ and 7.76 nm for $FAPbBr_3$[52]), the samples should display similar size-induced quantum confinement. Samples were prepared following recently published room-temperature procedures (Supplementary Note 1)[39]. Compared to $CsPbBr_3$, the PL peak center of a $FAPbBr_3$ QD is redshifted by 72 meV and the PL peak width (full width at half maximum; FWHM) is increased by 13 meV. These spectral trends are consistent across a larger single-QD dataset (Supplementary Fig. 2) and with ensemble PL measurements (Supplementary Fig. 3). The red-shift is associated with the straightened and stretched Pb-Br-Pb bonding (Fig. 2a-d), and the increased peak width suggests stronger electron-phonon coupling in $FAPbBr_3$[26,48].

Moreover, we observed an exponential low-energy (Urbach) tail in single-particle PL spectra, see Fig. 2f. Across a large size range, the tails are consistently steeper for $CsPbBr_3$ than in comparably sized $FAPbBr_3$ QDs (Fig. 2g). As elucidated in a prior study[48], this suggests stronger exciton-phonon coupling in $FAPbBr_3$ QDs, consistent with PL phonon replica at cryogenic temperatures[53,54] and optical-pump–electron-diffraction-probe measurements[55]. This trend is also confirmed by our accompanying density functional theory (Supplementary Fig. 4) and ensemble PL studies (Supplementary Fig. 3). Urbach tails are associated with exciton-phonon coupling through the formation of localized states that result from thermal disorder and form the low-energy tail[32,34,36]. They thus also serve as a first indication of the hypothesized wavefunction confinement which may be enhanced in pseudo-cubic $FAPbBr_3$ QDs.

### Phonon-driven wavefunction localization
An understanding of electron-phonon coupling can be gleaned from finite-temperature ab-initio molecular-dynamics (AIMD) simulations based on density functional theory (DFT). We performed simulations for 3.6 nm large ABr-terminated $CsPbBr_3$ and $FAPbBr_3$ QD models at the Perdew-Burke-Ernzerhof level of theory (see "Methods")[56]. Previously, such a balance between computationally approaching experimental QD sizes yet ensuring a sufficiently high level of theory has successfully reproduced experimental results related to surface chemistry[57] and exciton-phonon interaction[7,26]. To study wavefunction confinement, we computed the wavefunction of the highest occupied molecular orbital (HOMO) which represents the hole wavefunction. Figure 3a displays representative snapshots of the HOMO wavefunctions at 10 K and 300 K. At low temperatures, the wavefunctions fully delocalize across the entire QD volume for both $CsPbBr_3$ and $FAPbBr_3$ QDs. At 300 K, however, the wavefunction strongly contracts and localizes in the $FAPbBr_3$ QD, while such a thermally activated wavefunction confinement is considerably weaker in the $CsPbBr_3$ QD.

Figure 3b shows the wavefunction sizes averaged across >120 snapshots (>12 ps; further details in Supplementary Note 7) in the $CsPbBr_3$ and $FAPbBr_3$ QDs along AIMD trajectories at 10, 100, and 300 K (further temperatures in Supplementary Fig. 5). In both materials, the wavefunction sizes remain between 1.6 and 1.8 nm at 10 and

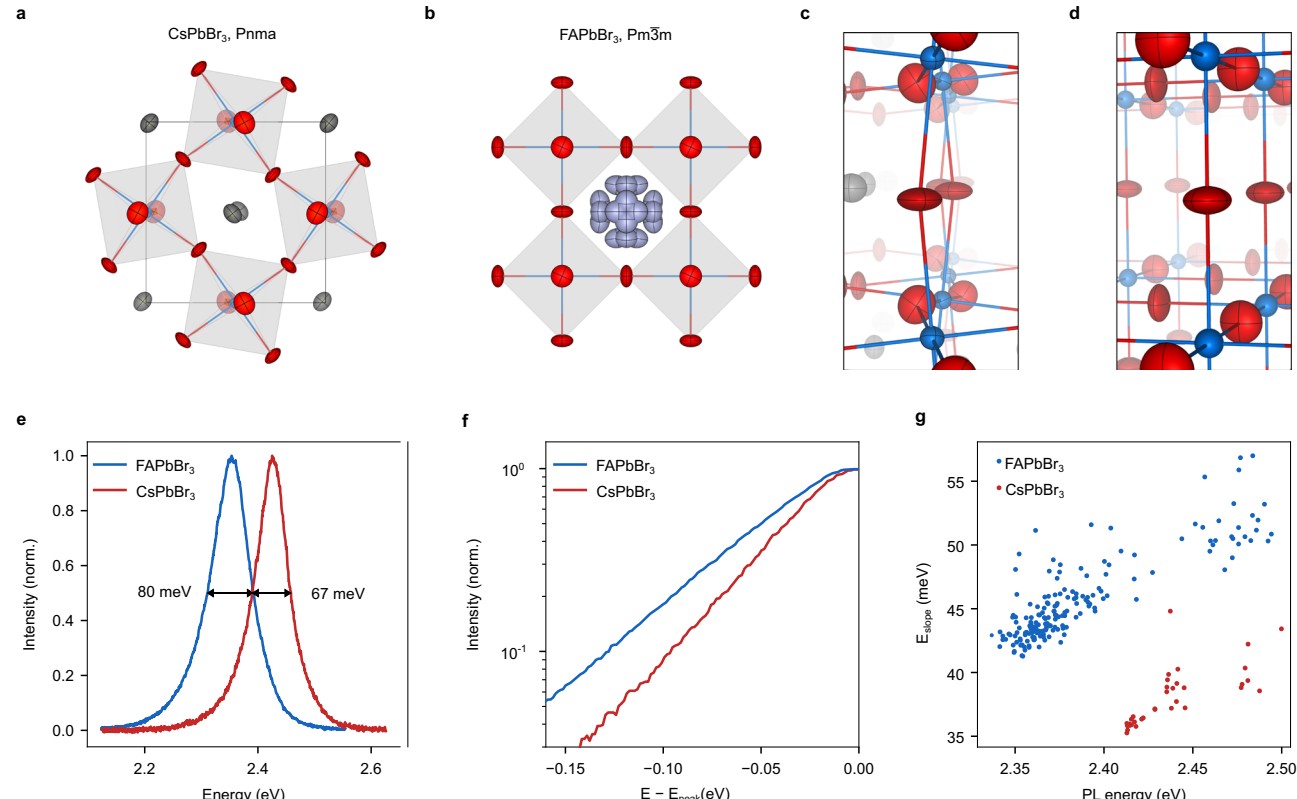

**Fig. 2 | Effect of the A-site cation on spectral features of perovskite QDs.** Room-temperature crystal structures of CsPbBr$_3$ (**a**) and FAPbBr$_3$ (**b**) obtained by single-crystal X-ray diffraction (red: Br, blue: Pb, grey: Cs, purple: N, black: C; H omitted for clarity). Magnified view of the Pb-Br-Pb bond and displacement ellipsoids in the crystal structures of CsPbBr$_3$ (**c**) and FAPbBr$_3$ (**d**). The FA cation was omitted for clarity. **e** Photoluminescence spectra of single FAPbBr$_3$ and CsPbBr$_3$ QDs with similar size (both 9.9(1.2) nm) displaying a spectral redshift and spectral broadening upon replacing Cs by FA. **f** Red tail of the PL spectra from single FAPbBr$_3$ and CsPbBr$_3$ QDs, referenced to their PL center energy $E_{peak}$. **g** Slope energy ($E_{slope}$) as a function of PL center obtained by fitting eq (S1) to the red tail of the PL spectra of single FAPbBr$_3$ and CsPbBr$_3$ QDs of various sizes.

100 K, limited by QD size. Above 100 K, the wavefunction size of the FAPbBr$_3$ QD decreases to just $0.93 \pm 0.17$ nm, whilst the wavefunction size of the CsPbBr$_3$ QD moderately decreases to $1.40 \pm 0.05$ nm. This observation is consistent across several QD diameters up to 5.4 nm (Supplementary Fig. 6-8). While the bandgap in both systems is still influenced by the QD size (Supplementary Fig. 9), the wavefunction size for FAPbBr$_3$ QDs at 300 K is largely limited by the disorder-induced wavefunction localization (Supplementary Fig. 6). In contrast, CsPbBr$_3$ retains a QD-size dependent wavefunction extension. The electron wavefunction, represented by the lowest unoccupied molecular orbital (LUMO), experiences similar wavefunction localization as the HOMO (Supplementary Note 8, Supplementary Fig. 10, 11). Finally, temperature-induced localization even occurs in the respective bulk materials, specifically for FAPbBr$_3$ (Supplementary Fig. 11, 12). Although bulk crystals will not experience Auger-Meitner relaxation due to vanishing inter-exciton coupling, such localization effects reduce radiative and non-radiative rates and limit carrier mobilities in lead halide perovskites[45,58–60]. Such a systematically observed wavefunction localization across model sizes ranging from 1.8 nm QDs to the bulk alongside the qualitative agreement in key experimental observations linked to exciton-phonon coupling (Supplementary Fig. 3,4)[7,26], affirms that such wavefunction localization also occurs in the perovskite QD sizes accessed experimentally (vide infra).

Importantly, the wavefunction localization is dynamic. Both the size and location inside the QD vary between snapshots along the MD trajectory conveying a transient localization behavior of the wavefunction (Supplementary Videos, Supplementary Fig. 13). Moreover, the time-averaged wavefunction, integrated across the entire AIMD trajectories (Supplementary Fig. 14), smoothly covers the entire QD volume which excludes static contributions to the localization. The highly dynamic wavefunction as well as the temperature-dependence suggest that anharmonic phonons are the primary cause for the transient structural disorder and the associated wavefunction localization. This mechanism is reinforced by temperature-dependent radial and angular distribution functions that support the existence of temperature-induced atomic displacement from their average position (Supplementary Figs. 15-17). Moreover, the structural disorder caused by these atomic displacements is discernible from temperature-dependent spatial correlation functions of the PbBr$_3^-$ framework (Supplementary Fig. 18). Finally, the correlation of temperature-induced structural disorder and wavefunction localization with the population of phonon modes underpins allocating the disorder to phonons (Supplementary Fig. 19).

Further insight into the phonon-driven wavefunction localization is provided by the autocorrelation function of the HOMO wavefunction coefficients which describes the time evolution of the wavefunction (Supplementary Note 7, Supplementary Equation 9). Figure 3c depicts the normalized wavefunction autocorrelation functions at 300 K exhibiting decays within hundreds of femtoseconds that envelop oscillations with a period of roughly 200 fs. Notably, the loss of correlation is significantly faster for the FAPbBr$_3$ QD than for the CsPbBr$_3$ QD[7]. This observation resembles the bandgap-energy autocorrelations, which have previously been assessed to derive the phonon-driven electronic dephasing in these and other semiconductor materials[26,61,62].

The power spectra of the wavefunction autocorrelation (Supplementary Note 7) in the inset in Fig. 3c reveal the dominant vibrational features driving the wavefunction localization in both systems. While

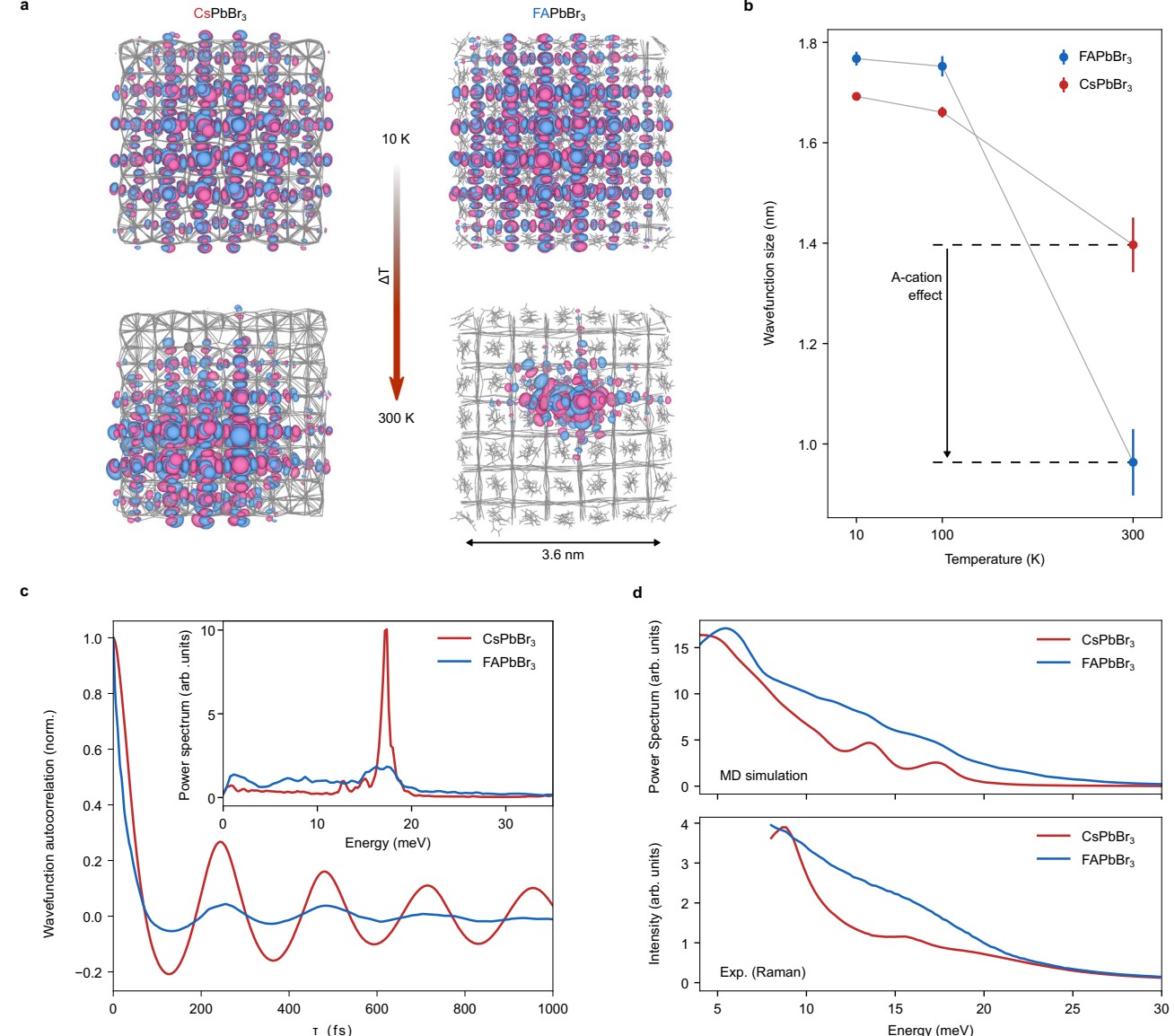

**Fig. 3 | Temperature-induced dynamic wavefunction localization and structural origin. a** Representative snapshots of HOMO wavefunctions obtained from ab-initio molecular dynamics (AIMD) simulations of CsPbBr$_3$ and FAPbBr$_3$ QDs with edge lengths of 3.6 nm at 10 and 300 K. **b** Average HOMO wavefunction size from AIMD trajectories between 10 and 300 K. Error bars indicate 95% confidence intervals. **c** Autocorrelation function of the HOMO wavefunction obtained by AIMD at 300 K normalized to its value at zero delay. Inset: Spectral density of the wavefunction autocorrelation. **d** Vibrational spectral density from AIMD simulations at 300 K (top) and experimental Raman spectra (bottom) of CsPbBr$_3$ and FAPbBr$_3$ QDs at 300 K.

the wavefunction-phonon coupling strength in CsPbBr$_3$ essentially concentrates within a single peak at 17 meV, the coupling strength in FAPbBr$_3$ QD additionally derives from a broad range of low-energy phonons. The population of these modes is confirmed by vibrational spectral density at these energies in our AIMD simulations as well as experimental Raman spectra with good agreement between the two methods (Fig. 3d). The broad signal in FAPbBr$_3$, contrary to the narrow peak in CsPbBr$_3$, attests a high degree of anharmonicity. Whilst the 17 meV peak is associated with a stretching mode along the Pb-Br bonds, the highly anharmonic tilting of PbBr$_6$ octahedra contributes to the lower-energy modes[53,55]. Peculiar to average cubic perovskites like FAPbBr$_3$, this anharmonic motion induces local and transient symmetry breaking, prompting their labelling as pseudo-cubic[63–66]. Because it also causes a significant bandgap renormalization[63], the local symmetry breaking is expected to affect the wavefunction localization of the excitons strongly. Consequently, tuning between the orthorhombic and pseudo-cubic structure through the A-site cation

allows us to manipulate the disorder potential and the resulting wavefunction localization. The local symmetry breaking through octahedral tilting and the influence of the A-site cation on the structural disorder are apparent in X-ray diffuse scattering experiments of FAPbBr$_3$ and CsPbBr$_3$ single crystals (Supplementary Fig. 20)[66–68]. Based on structural similarities[69], we expect other pseudo-cubic perovskites like methylammonium lead bromide (MAPbBr$_3$) or aziridinium lead bromide (AZPbBr$_3$) to exhibit similarly pronounced wavefunction localization.

## Purified single-photon emission

We now assess the single-photon purity of similarly sized (9.9(1.2) nm) CsPbBr$_3$ QDs and FAPbBr$_3$ QDs in single-particle PL measurements. In CsPbBr$_3$ QDs, $g^2(0)$ values are strongly size-dependent and increase drastically with increasing QD size[18]. Fig. 4a shows a representative second-order photon-photon correlation ($g^2(\tau)$) of a CsPbBr$_3$ QD. The incomplete anti-bunching of $g^2(0) = 0.29$ translates into a single-

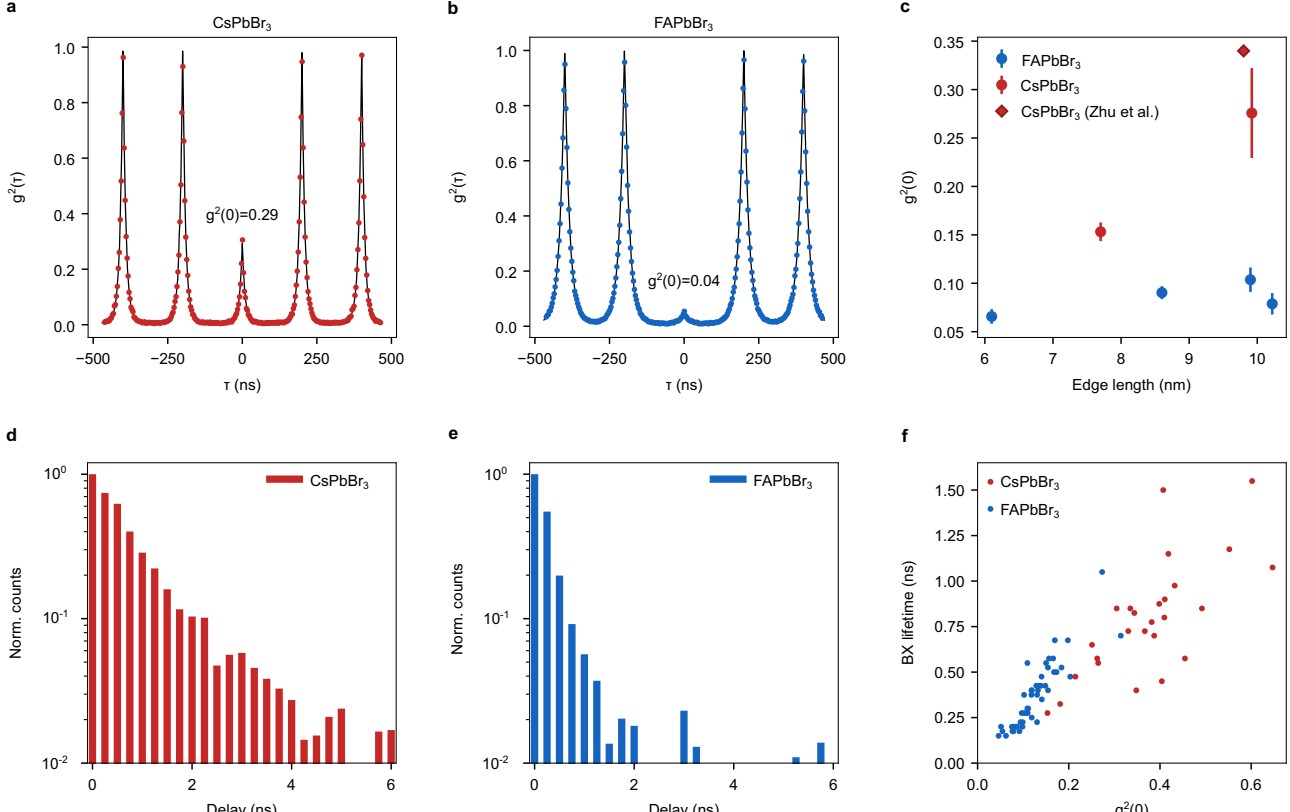

**Fig. 4 | Heralded single-particle PL spectroscopy at 300 K reveals suppressed multi-photon emission.** Second-order photon-photon correlation functions ($g^2(\tau)$) of similarly size-confined CsPbBr$_3$ (**a**) and FAPbBr$_3$ (**b**) QDs with relative sizes of 1.78 and 1.42 times the Bohr diameter $d_B$. A strong anti-bunching dip in the coincidences at zero delay time ($g^2(0)$) indicates a high single-photon purity and a low biexciton quantum yield. **c** Size dependence of $g^2(0)$ for various APbX$_3$ (A=Cs/ FA, X=Br/I) QDs. The relative size is expressed in terms of $d_B$. Error bars indicate 95% confidence intervals. Datapoints for CsPbX$_3$ were extracted from ref. [18]. Single-particle biexciton PL decay traces obtained for similarly size-confined CsPbBr$_3$ (**d**) and FAPbBr$_3$ (**e**) QDs. **f** Single-particle biexciton (BX) lifetimes as a function of $g^2(0)$. Enhanced single-photon purity (equivalent to lower $g^2(0)$) correlates with smaller biexciton lifetimes, i.e., quenched multi-exciton emission.

photon purity of only 71%, with a high probability of biexciton emission. This observation is consistent with the behaviour of traditional semiconductors for which QD-size confinement is the major knob for tuning multiexciton quenching via Auger-Meitner recombination. When size confinement is progressively lost upon increasing the QD size, so is the Coulomb interaction mediating the Auger-Meitner recombination of multi-excitons. Consequently, for large QDs with a size exceeding the exciton Bohr diameter, the radiative decay of multi-excitons may increasingly become a competitive channel, hereby compromising the single-photon purity. Several reports have indeed shown that large CsPbBr$_3$ QDs lose their characteristic anti-bunching due to multi-photon emission from biexcitons[18,70,71].

In contrast, despite a similarly weak QD-size confinement, a representative FAPbBr$_3$ QD achieves a drastically higher single-photon purity of 96% ($g^2(0) = 0.04$) (Fig. 4b). Figure 4c shows the size dependence of the single-photon purity including data from >150 single QDs as well as previously published results from our laboratory[18]. FAPbBr$_3$ QDs exhibit systematically smaller $g^2(0)$ than comparably size-confined CsPbBr$_3$ QDs supporting the postulated enhancement of quantum confinement in the cubic FAPbBr$_3$ QDs. In fact, CsPbBr$_3$ QDs with an edge length of 17.2 nm are only weakly anti-bunched ($g^2(0) = 0.3$-0.7), while the $g^2(0)$ from FAPbBr$_3$ QDs of the same size remain well below 0.5 (Supplementary Fig. 21). Although the concept of Bohr diameters is not strictly valid at room temperature for materials exhibiting strong exciton-phonon coupling, we additionally corroborated the A-site-cation dependent single-photon purity for samples with identical sizes normalized by Bohr diameters (Supplementary Fig. 22).

In a low-excitation-density regime, $g^2(0)$ corresponds to the ratio of the biexciton QY to the exciton QY[19]. Given identical exciton QYs in CsPbBr$_3$ and FAPbBr$_3$ QDs (88% and 89%, respectively), our experiments point to quenched biexciton QYs in FAPbBr$_3$ QDs either (i) due to a speed-up of non-radiative Auger-Meitner recombination of the biexciton and/or (ii) due to slower radiative decay of the biexciton. To disentangle the two contributions, we selectively probe the biexciton emission using heralded single-particle spectroscopy (Supplementary Note 5)[72]. Fig. 4d displays a representative biexciton PL decay trace of a CsPbBr$_3$ QD with a biexciton lifetime of 0.8 ns. In a representative FAPbBr$_3$ QD, the biexciton PL decays significantly faster with a lifetime of just 0.4 ns (Fig. 4e) despite a slower radiative decay of the exciton (Supplementary Fig. 23). Furthermore, the statistics from 77 QDs show that, while both mechanism (i) and (ii) contribute to quenched biexciton QYs, mechanism (i), i.e., an accelerated non-radiative recombination, is the main lever in quenching biexcitons and enhancing single-photon purity in FAPbBr$_3$ QDs (Fig. 4f).

We introduced the phonon-induced wavefunction localization as a temperature-dependent effect in the previous section. Therefore, we assessed the temperature dependence of the photon anti-bunching ($g^2(0)$) for FAPbBr$_3$ and CsPbBr$_3$ QDs of similar sizes (Supplementary Figs. 21, 24, 25). Second-order correlation functions are not anti-bunched at 4 K for single QDs of either material, attesting high biexciton PLQYs at cryogenic temperature. Matching the temperature dependence of the wavefunction localization (Fig. 3b), the single-photon purity increases with increasing temperature in both materials (Supplementary Figs. 21, 24). It is noteworthy that this temperature dependence could in parts be linked to single-photon superradiance, a

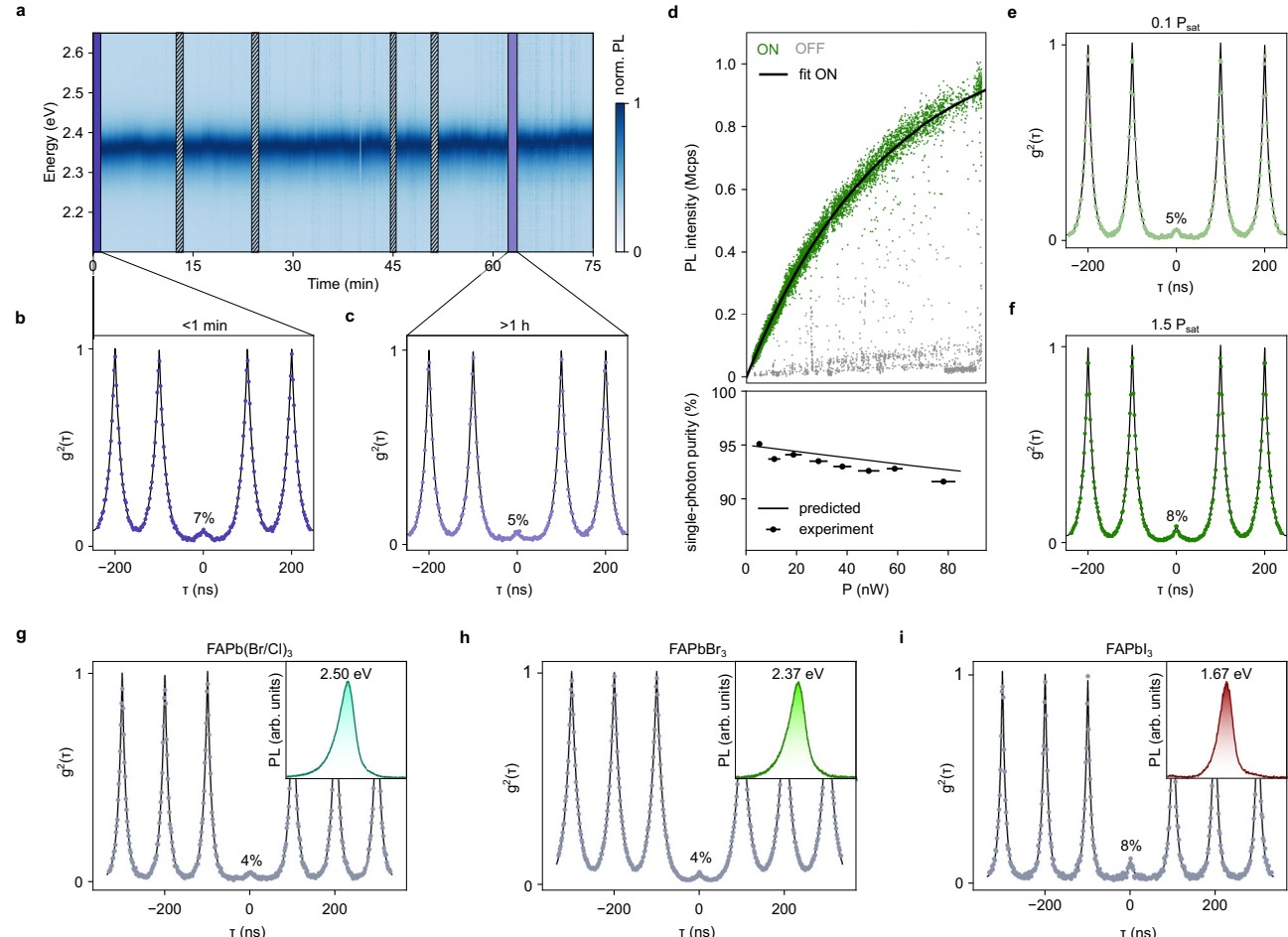

**Fig. 5 | FAPbX$_3$ (X=Cl,Br,I) QDs as stable, bright and spectrally tuneable single-photon sources. a** PL spectra series of a highly photostable FAPbBr$_3$ QD during >1 h of continuous irradiation. Shaded areas correspond to coincidence measurements (g$^2$(τ)). Measured (green data points) and fitted (black lines) g$^2$(τ) of the QD before (**b**) and after (**c**) irradiation for 1 h. **d** Top: Excitation-power dependence of the PL intensity of an individual FAPbBr$_3$ QD pumped at a repetition rate of 10 MHz. Datapoints corresponding to the QDs bright ON state (blue dots) were fitted by a saturation model (red line) to extract a maximum brightness of 1.1 Mcps and a saturation power density of 70 W/cm². Bottom: Single-photon purity as a function of excitation power. The grey curve indicates the ideal and background-free behaviour of an ideal QD with a biexciton QY of 0.05 times the exciton QY. Error bars indicate the integrated range of excitation power. Measured (blue data points) and fitted (black lines) g$^2$(τ) of the QD recorded at 10% (**e**) and 150% (**f**) of the saturation power P$_{sat}$. Measured (blue data points) and fitted (black lines) g$^2$(τ) and PL spectrum (inset) of (**g**) a FAPb(Br/Cl)$_3$ QD emitting blue single photons (PL center at 2.50 eV, 495 nm), (**h**) a FAPbBr$_3$ QD emitting green single photons (PL center at 2.36 eV, 525 nm) and (**i**) a FAPbI$_3$ QD emitting red single photons (PL center at 1.67 eV, 742 nm).

process associated with a wavefunction delocalization that causes a rapid radiative decay and that is lost at higher temperatures due to thermal disorder[7], or to Auger-Meitner recombination intrinsically requiring thermal activation[73,74]. Nevertheless, a stronger increase of single-photon purity observed in FAPbBr$_3$ demonstrates that disorder-induced wavefunction localization plays a key role (Supplementary Figs. 21, 22, 24). These temperature and composition trends are also followed by the increasing PL peak width, an indicator for exciton-phonon coupling, and the increasing PL lifetime (Supplementary Fig. 3), which was recently linked with wavefunctions localized through vibrations[58,59]. All these effects reflect the disorder-induced wavefunction localization.

Our AIMD simulations uncovered that disorder-induced wavefunction localization is related to octahedral tilting causing local symmetry breaking in pseudo-cubic perovskites. Pseudo-cubic APbBr$_3$ QDs can also be obtained for other organic A-site cations, such as methylammonium (MA) and aziridinium (AZ). Indeed, weakly size-confined MAPbBr$_3$ QDs and AZPbBr$_3$ QDs exhibit an average single-photon purity of 90(5)% and 89(3)%, comparable to FAPbBr$_3$ QDs of similar size confinement (Supplementary Figs. 26, 27)[39,69], attesting the universality of the proposed phonon-mediated mechanism.

## FAPbX$_3$ QDs as single-photon sources

Efficiently quenching multi-excitons via phonon-driven wavefunction confinement opens a new and straightforward avenue toward (i) stable, (ii) bright, and (iii) color-tunable single-photon sources - all enabled by the here-introduced route to circumvent the previous need for QD downsizing to achieve high single-photon purity.

Stability issues are predominantly associated with the QD surface, especially for smaller QDs with high surface-to-volume ratios, which exhibit notably poor photostability. Photodegradation has been attributed to surface-matrix reactions, eventually leading to a shrinkage of the QD core that manifests itself in a dynamical spectral blue-shift and a reduction in PL intensity[75]. Indeed, an individual QD from a large FAPbBr$_3$ QD sample (10.1(9) nm) exhibits extreme photostability with spectrally stable emission during continuous irradiation for 75 minutes, see Fig. 5a. Figure 5b, c display the g$^2$(τ) measured before and after irradiation of 1 hour, respectively. Strong anti-bunching with g$^2$(0) of 7% and 5% conveys that single-photon purity does not deteriorate over the course of continuous irradiation for 1 hour. Moreover, the QD displays only very weak blinking, on par with sophisticated core-shell structures[9,10] and remains largely unchanged. Before irradiation, we observe that the QD spends 98% in its bright (ON) state,

while after 1 hour of continuous irradiation an ON fraction of 94% is retained (Supplementary Fig. 28).

Several quantum-engineering applications based on room-temperature single-photon emitters specifically demand bright emitters. For example, quantum key distribution generally requires emitters with high brightness at a single-photon purity >90%[4]. Fig. 5d displays the power-dependent PL intensity (top panel) and single-photon purity (bottom panel) of an individual large and weakly confined QD at a laser repetition rate of 10 MHz. We extract a saturation power density $P_{sat}$ of 70 W/cm$^2$ and a maximum brightness of 1.1 Mcps suggesting that the radiative rate limits the brightness (details in Supporting Information). At a low excitation power (0.1 $P_{sat}$), strong anti-bunching attests a single-photon purity of 95% (Fig. 5e). An increase of the excitation to 1.5 $P_{sat}$ only reduces the single-photon purity to 92% (Fig. 5f). In quantum emitters such as defects or molecules, single-photon purity in saturation can be deteriorated by strong background emission at high laser power. However, large perovskite QDs do not suffer from this limitation owing to their orders of magnitude larger absorption cross sections. This is illustrated by the agreement between the measured power dependence of the single-photon purity and the background-free prediction for a biexciton PLQY of 5% of the exciton PLQY (bottom panel; Fig. 5d).

Lastly, extending high single-photon purity from small to large FAPbX$_3$ QDs also allows spectral fine-tuning of single-photon emission. Together with the already available compositional tuning through the halide identity, FAPbX$_3$ QDs can now be precision-engineered to deliver high single-photon purity across the entire visible range, for example blue-emitting FAPb(Br/Cl)$_3$ (Fig. 5g; 9.2(8) nm edge length), green-emitting FAPbBr$_3$ (Fig. 5h, 10.2(1.2) nm edge length), and far-red emitting FAPbI$_3$ (Fig. 5i, 11.3(1.3) nm edge length). All QDs display strong anti-bunching certifying single-photon purities >90%. Blue-emitting single-photon sources receive increasing interest for underwater communication[76,77]. Meanwhile, reaching the far-red and near-infrared spectrum with FAPbI$_3$ QDs is particularly relevant due to reduced losses in fiber-optic communication[2] and biological tissue[78].

In summary, we discovered a strategy to enhance quantum confinement in colloidal QDs and improve their single-photon purity without the need for QD downsizing. Caused by anharmonic vibrations that couple to electronic degrees of freedom, dynamic disorder induces wavefunction confinement which increases with temperature and can be tuned through the control of crystal vibrations. In our implementation, transient wavefunction confinement is realized by deploying organic A-site cations in lead halide perovskite QDs yielding pseudo-cubic structures with a tendency towards dynamic local-symmetry breaking. Phonon-induced wavefunction localization presents a counterintuitive but beneficial effect of strong exciton-phonon coupling, contrasting the typically negative connotation of such coupling in optoelectronic materials. Owing to the resulting purified single-photon emission in large QDs, we can report competitive brightness, photostability and single-photon purity while retaining the broad spectral tunability that make colloidal QDs stand out amongst room-temperature single-photon sources. These characteristics are expected to further improve under more favourable conditions such as resonant excitation[79,80], temporal and spectral filtering[7,80,81], or integration into optical microcavities and nano antennae[82,83]. Taken together, large colloidal organic-inorganic perovskite QDs are a versatile and scalable platform for room-temperature single-photon sources. Considering the ubiquity of electron-phonon coupling, we are confident that actively tuning this interaction provides opportunities for functional material engineering beyond perovskite QDs.

## Methods

### Representation of errors
Throughout this work, standard deviations are reported in the text in rounded brackets, while 95% confidence intervals are reported after a ±-sign. Error bars in Figs. 3 and 4 represent 95% confidence intervals, as stated in the caption.

### Single-crystal X-ray diffraction
Single-crystal growth was conducted using the inverse-temperature crystallization method for FAPbBr$_3$ and CsPbBr$_3$. Detailed descriptions are found Supplementary Note 3.

Single-crystal X-ray diffraction data for crystal-structure determination were collected on a Rigaku Synergy S diffractometer equipped with microfocus Mo K-alpha X-ray source, a Dectris Pilatus 300 K hybrid pixel detector, and an Oxford Cryostream 800 for temperature control. Diffraction images were acquired every 0.2° of rotation, during continuous rotation of the crystal around the phi axis. The resulting data was processed with the software CrysAlisPro (Rigaku Oxford Diffraction) for indexing, instrumental model refinement, and intensities integration. Descriptions of structure solutions and refinement are provided in the Supporting Information. Results are detailed in Tables S1-3, included as Supporting Data, and available in the CSD and ICSD databases under deposition numbers 2407691 (FAPbBr$_3$) and 2407795 (CsPbBr$_3$).

For X-ray total scattering, selected specimens were mounted on a MiTeGen kapton loop using inert NVH high viscosity oil. Diffraction data was collected by a Rigaku Synergy S diffractometer equipped with a Pilatus 300 K hybrid pixel detector, Mo K-alpha microfocus source, and an Oxford Cryostream system set at a temperature of 300 K. Further details regarding data collection, processing and analysis are provided in Supplementary Note 3.

### Colloidal quantum dot synthesis
FAPbBr$_3$, CsPbBr$_3$, MAPbBr$_3$ and FAPb(Br/Cl)$_3$ QDs were synthesized following the methods described in ref. 39. FAPbI$_3$ QDs were prepared according to ref. 84. The synthesis of AZPbBr$_3$ QDs followed ref. 69. Detailed descriptions of the syntheses are provided in Supplementary Note 1.

### Transmission electron microscopy
Transmission electron microscopy (TEM) images were acquired either using a Hitachi HT7700 microscope operated at 100 kV or using a JEOL Model JEM-2200FS microscope operated at 200 kV. Scanning transmission electron microscopy (STEM) images were collected using a JEOL Model JEM-2200FS microscope operated at 200 kV. Particle sizes were determined using ImageJ.

### Raman spectroscopy
Raman spectroscopy was performed using a confocal Raman microscope (Horiba LabRAM HR Evolution). FAPbBr$_3$ and CsPbBr$_3$ QD thin films were prepared by dropcasting respective QD colloids under ambient conditions onto a glass microscope slide, cleaned previously with ethanol. A 785 nm cw laser (Toptica XTRA II; about 30 mW) was focused onto the sample by an objective (Olympus MPlanN; NA = 0.9, 100x magnification). The scattered light was collected by the same objective, Rayleigh-scattered light removed by a 785 nm longpass filter, and the remaining Raman-scattered light spectrally dispersed onto a CCD camera (Horiba Synapse) using a grating (300 g/mm, blazed at 600 nm). All Raman spectra are reported after wavenumber calibration (using a Si wafer) and correction for the spectral response of the detector. The samples were stable throughout the acquisition time of ~100 s and no sample damage was observed due to the Raman excitation.

### Variable-temperature time-integrated and time-resolved PL measurements
CsPbBr$_3$ and FAPbBr$_3$ QD thin films were prepared via spincoating onto Si substrates with a 2 μm thick SiO$_2$ surface oxide layer. The samples were mounted in a liquid-helium closed-cycle cold-finger cryostat

(ARS; DE204AE; sample in vacuum) and probed by a PL spectrometer (Picoquant; FluoTime 300) using a pulsed 354.3 nm laser for excitation and a photomultiplier tube for detection. The laser power was adjusted to ensure low excitation densities, with less than one exciton per QD on average. Time-integrated PL spectra were acquired using a monochromator in the detection path, employing a bandwidth of 0.7 nm and 80 MHz repetition rate. Time-resolved PL decays were acquired at the PL maximum, with an emission bandwidth of 27 nm, to average out spectral dynamics, and employing a repetition rate of 2.5 MHz. To record temperature-dependent PL characteristics, PL spectra and decays were acquired at several temperatures between 17 K to 296 K. All PL spectra were corrected for dark counts and the spectrally varying detector sensitivity. All time-resolved PL decays were corrected for dark counts. Both PL spectra and time-resolved PL decays were normalized to the peak counts.

### Single-particle optical spectroscopy

Samples for single-particle optical spectroscopy were prepared in nitrogen filled gloveboxes using anhydrous solvents. As-synthesized QD samples were diluted in multiple steps by three to five orders of magnitude in cyclohexane, toluene, or octane and spin-coated (100 μL, 150 revolutions per second, 60 s) onto clean cover glasses or, prior to spin-coating, diluted by additional one to two orders of magnitude in 3 mass% solutions of polymers (SEBS in cyclohexane or polystyrene in toluene) for additional protection from moisture and air. The resulting sparse QD films were placed in a nitrogen filled sample holder or measured under ambient conditions. Full details of the preparation of single-dot samples are provided in Supplementary Note 4.

Conventional single-particle spectroscopy was performed with a custom-built μPL setup. A 405 nm pulsed excitation laser (PicoQuant, 1–10 MHz repetition rate, <50 ps pulse width, <100 W/cm²) is focussed with an oil immersion objective ($1/e^2 = 1\,\mu m$, 1.3 NA) onto the sample that is mounted on XYZ translational stages (SmarAct, <1 nm resolution). Collected by the same objective, the light emitted from single QDs is passed through a dichroic mirror as well as a long-pass filter (both 450 nm cut-on wavelengths). The filtered light is either sent to a monochromator coupled to an EMCCD (Princeton Instruments, one frame per second) or to a Hanbury-Brown and Twiss setup consisting of a 50:50 beam splitter, two avalanche photodiodes (Excelitas, 250 ps time resolution), a time-correlated single-photon counting module (PicoQuant, HydraHarp) and a short-pass filter (cut-off wavelength 750 nm). PL spectra were recorded before and after measurements in the HBT setup to ensure that QDs did not blueshift under illumination. The raw data was processed in custom python codes or using SymPhoTime (PicoQuant). Complete details of the data analysis are provided in Supplementary Note 4.

Heralded single-particle spectroscopy utilized a SPAD array spectrometer previously described in refs. [72,85] and built around a commercial inverted microscope (Eclipse Ti-U, Nikon). Excitation light from a pulsed laser source (470 nm, 5 MHz, LDH-P-C-470B, PicoQuant) is focused onto the sample with an oil immersion objective (×100, 1.3 NA, Nikon). The emitted light is collected by the same objective and passed through a dichroic mirror (FF484-FDi02-t3, Semrock) and a long-pass filter (BLP01-473R, Semrock). The magnified image plane (×150) serves as the input for a Czerny-Turner spectrometer that consists of a 4-f system (AC254-300-A-ML and AC254-100-A-ML, Thorlabs) with a blazed grating (53-*-201R, Richardson) at the Fourier plane. At the output image plane of the spectrometer, a 512-pixel on-chip linear SPAD array is placed. Photon pairs following the same excitation pulse are post-selected as heralded events and energy- as well as time-tagged to reconstruct spectra and PL decays of the first (biexciton) and second photon (exciton). Further details on the SPAD array spectrometer and data processing are found in Supplementary Note 5.

Single-particle spectroscopy at cryogenic temperatures was performed in a custom-built μPL setup equipped with an evacuated liquid-helium closed-loop cryostat (Montana Instruments). Single QDs were excited using a tuneable fs pulsed laser (set to 480 nm and 80 MHz, Toptica) that is fiber coupled and focused on the sample by a microscope objective (NA = 0.8, 100x, dry). PL from single QDs was collected by the same objective, passed through a long-pass filter (cut-off 500 nm), and sent either into the monochromator coupled to back-illuminated CCD camera (Princeton Instruments) to record the spectra or towards the HBT setup with a 50:50 beam splitter and two SPDs (ca. 50 ps time resolution) to record the second-order intensity correlation. Signals from SPDs were fed into the TCSPC module (Picoharp, PicoQuant). PL spectra were recorded with a grating of 300 lines per mm, blaze at 500 nm, giving a spectral resolution of around 1 meV. Further experimental details and data processing are found in Supplementary Note 6.

### AIMD simulations

Finite-size ab-initio molecular-dynamics (AIMD) simulations were performed using density functional theory (DFT). Previously published models[26] of AX-terminated cuboidal $CsPbBr_3$ and $FAPbBr_3$ QDs with truncated corners and approximate edge lengths of 1.2, 2.4, 3.0, 3.6 and 5.4 nm were placed in vacuum cell (at least 1 nm on each cell). We used the Quickstep module in CP2K employing Gaussian and plane waves with a plane-wave cutoff of 280 Ry[86], DZVP-MOLOPT basis sets[87], Goedecker-Teter-Hutter pseudopotentials[88] and Perdew-Burke-Ernzerhof exchange-correlation functionals[56]. Molecular dynamics (MD) simulations in the NVT ensemble employed timesteps of 1 or 10 fs and a canonical sampling through velocity rescaling thermostat[89] with a time constant of 15 or 250 fs. Trajectories with a duration of at least 12 ps were produced after equilibration at the respective temperature for 6 ps. To analyse the wavefunction confinement in real space, cube files for the highest occupied molecular orbital (HOMO) wavefunctions were extracted every 0.1 ps (>120 snapshots). The analyses of the trajectories yielding wavefunction sizes and descriptors of disorder are detailed in the Supporting Information. MD simulations in the NVE ensemble were performed after thermal equilibration of at least 6 ps in the NVT ensemble. Trajectories with timesteps of 1 fs and a duration of at least 10 ps were prepared for to analyse nuclear vibrations and extract wavefunction dynamics through the molecular orbital coefficients, details of which are provided in the Supplementary Note 7.

Periodic-boundary AIMD simulations were performed using DFT. We used the Quickstep module in CP2K employing Gaussian and plane waves with a plane-wave cutoff of 280 Ry[86], DZVP-MOLOPT basis sets[87], Goedecker-Teter-Hutter pseudopotentials[88] and Perdew-Burke-Ernzerhof exchange-correlation functionals[56]. As starting structures of MD simulations, 4x4x4 orthorhombic supercells of $FAPbBr_3$ and $CsPbBr_3$ were constructed. MD simulations in the NPT ensemble employed time steps of 1 or 10 fs, a canonical sampling through velocity rescaling thermostat[89] with a time constant of 15 fs, and a barostat with a time constant of 1000 fs. Trajectories with a duration of at least 7 ps were produced after equilibration at the respective temperature for at least 3 ps. To analyse the wavefunction confinement, cube files for the HOMO and LUMO wavefunctions were extracted every 0.1 ps and their inverse participation ratio was calculated (details in Supplementary Note 7, 8).

### Data availability

All data supporting the findings in this study is available through Zenodo (doi:10.5281/zenodo.10977759). Additional details on computational methods, quantum dot synthesis, Raman spectroscopy, X-ray diffraction, and single-dot experiments (PDF). Videos of the highest occupied molecular orbital wavefunction densities in molecular dynamics simulations at 300 K of a 3.6 nm $FAPbBr_3$ QD (Supplementary Video 1, MP4), a 3.6 nm $CsPbBr_3$ QD (Supplementary

Video 2, MP4), a 5.4 nm $CsPbBr_3$ QD (Supplementary Video 3, MP4), bulk $CsPbBr_3$ (Supplementary Video 4, MP4), and bulk $FAPbBr_3$ (Supplementary Video 5, MP4). Videos of the lowest unoccupied molecular orbital wavefunction densities in molecular dynamics simulations at 300 K of bulk $CsPbBr_3$ (Supplementary Video 6, MP4), bulk $FAPbBr_3$ (Supplementary Video 7, MP4), and a 3.6 nm $CsPbBr_3/CsCaBr_3$ core/shell QD (Supplementary Video 8, MP4). Crystal structure solutions of $CsPbBr_3$ (2407795, CIF) and $FAPbBr_3$ (2407691, CIF) single crystals.

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

## Acknowledgements

This work was financially supported by the Swiss National Science Foundation (SNSF) through NCCR Catalysis (grant number 180544, M.V.K.) and the SNSF grant 200021_188404 (M.V.K), by the Weizmann-ETH Zurich Bridge Program (M.V.K, G.R, N.Y., V.W., D.O.), by the

European Union's Horizon 2020 program through a FET Open research and innovation action (Grant Agreement No. 899141, "PoLLoC", M.V.K., G.R.), and by the Air Force Office of Scientific Research under award number FA8655-21-1-7013 (M.V.K., G.R.). The authors acknowledge support from the Swiss National Supercomputing Centre (CSCS; project ID s1230, N.Y., V.W.). T.K. acknowledges support from the ETH Zurich Postdoctoral Fellowship (grant number 23-2 FEL-021, T.K.).

## Author contributions

L.G.F. conducted room-temperature single-particle PL studies and performed and analyzed AIMD simulations of QDs. N.F., G.L. and M.K., supervised by D.O., performed heralded single-particle PL studies. T.K., C.Z. and L.G.F. conducted single-particle PL studies at cryogenic temperatures. N.Y., L.G.F. and V.W. performed and analyzed AIMD simulations of bulk materials. S.S. and S.C. prepared single crystals and conducted single-crystal X-ray scattering studies. S.C.B. acquired Raman spectra and performed variable-temperature ensemble PL studies. V.M., M.S., R.T., and M.I.B. synthesized QD solutions. G.R. and M.V.K. initiated and supervised the work. The manuscript was written by L.G.F. and G.R., with input from all authors. All authors discussed the results and commented on the manuscript.

## Competing interests

The authors declare no competing interests.
