## [Transparent Peer Review file · Nature Communications]

Phonon-driven wavefunction localization promotes room-temperature, pure single-photon emission in large hybrid lead halide perovskite quantum dots

Corresponding Author: Professor Maksym Kovalenko

Version 0:

Reviewer comments:

Reviewer #1

(Remarks to the Author)

In this manuscript, the authors propose a novel strategy to enhance quantum confinement in colloidal quantum dots (QDs) and improve their single-photon purity, eliminating the need for QD downsizing. Based on combined theoretical and spectroscopic approaches, it is revealed that the A-site cation in soft APbBr₃ colloidal QDs controls the phonon-induced localization of the exciton wavefunction. Detailed density functional theory (DFT) results demonstrate that the transient wavefunction confinement is realized via strong exciton-phonon coupling, contrasting the generally negative impact of such couplings in optoelectronic materials. In my opinion, this work presents a solid piece of work, which should be of interest to the community. It sheds new light on achieving bright, stable, and pure single-photon emission, while circumventing the optical stability issues associated with conventional QD downsizing. I would recommend its publication pending the authors' consideration of the following points:

1. In the computational section, have the selected methods undergone benchmark work to verify their applicability to the target system? Additionally, how are the weak interactions accounted for?
2. A key highlight of this work is the achievement of high-purity single-photon emission without the need for QD downsizing. What criteria determine the optimal QD size for this mechanism? Is there potential for this mechanism to be applicable to bulk materials as well?
3. As clarified in this manuscript, two types of QDs exhibit distinct phonon-driven wavefunction localization. Figure S3 in the Supporting Information presents the PL spectra of these two QDs as a function of temperature. I recommend that the authors conduct a quantitative analysis of the relationship between wavefunction localization and PL energy. Given that the PL energy shifts are generally regarded as directly related to wavefunction confinement, such a correlation could further validate the novel mechanism.
4. It is found that the loss of correlation is significantly faster for the FAPbBr QD than for the CsPbBr₃ QD. Generally, the phonon-driven electronic dephasing will severely limit the prospects of QDs as single-photon sources. How should this issue be comprehensively addressed? While it does not undermine the novelty of the work.

Reviewer #2

(Remarks to the Author)

The manuscript "Phonon-driven wavefunction localization promotes room temperature, pure single-photon emission in large hybrid lead halide perovskite quantum dots" by Feld and coworkers presents experimental evidences of the positive role from the dynamic disorder associated with the A-cation thermal motion on the single photon emission properties of halide perovskite quantum dots (QD). Namely, measurements of the second-order autocorrelation function of the photon arrival times ($g_2(t)$) at zero delay demonstrate single photon emission, at room temperature in CsPbBr₃ and FAPbBr₃ QDs, with single-photon purity of 71% and 96%, respectively. This is by itself an important results. The purity difference for the two compositions is then further rationalized using DFT-based Ab-Initio Molecular Dynamics (AIMD) simulations. The premise is that stronger confinement of the charge (as induced by reducing the QD size) improves single-photon purity (which I take as given, as I am not an expert myself in this specific field). From this premise, the authors provide convincing evidences from AIMD that the thermal motion in QD and related structural dynamic disorder localizes the electron/hole wavefunctions, ultimately acting as an additional confinement mechanism. Furthermore, they show that wavefunction localization is stronger in FAPbBr₃, as due to its specific dynamics.

Thought I am not an expert in the field of single-photon emission, I enjoyed reading this manuscript, its shape well suiting the broad audience of the journal. The results are overall convincing and represent to my understanding a step forward in the current understanding of the problem. However, before recommending its publication on Nature Communication, I believe the authors should try to clarify an important point.

As stated above, I like the message of the positive role of dynamic disorder on the single particle wavefunction confinement. Actually a similar mechanism was proposed long time ago by Wang (Nano Lett. 15, 248) and De Angelis (PCCP, 17,9394) to explain the reduced electron/hole recombination in 3D bulk halide perovskites. This however does not reduce the originality of the present work, as the charge localization is discussed here in relation to single photon-emission. However, the authors report indicative measurements of the wavefunction spread in Figure 3a,b, which set around 1-2 nm. On the other hand, when introducing the idea of spatial confinement from the exciton extension itself, in contrast to the QD size, the authors mention radii, the authors mention Bohr radii for the exciton of 6.16 nm and 7.76 nm, for CsPbBr₃ and FAPbBr₃, respectively. Even more, the Bohr radius is smaller for CsPbBr₃, while the results from AIMD point out stronger confinement from FAPbBr₃. This discrepancy somehow cast some shadow the full argument of the charge localization. Actually, it gives the impression that one compares “apples” with “pears”. Could the authors comment a bit on this? Why this discrepancy?

In addition, I take the occasion to rise few additional remarks, which are absolutely secondary but may still add something to the text.

1) I find the red signal in Figure 3 unexpectedly neat. For the sake of reference, the power spectrum of the band gap at 300 K of bulk halide perovskites was reported recently in Adv. Opt. Mater., 2301105, 2023, showing very messy signals in the frequency domain. Notice also that such clean signal somehow contrasts a lot with the importance of phonon anharmonicity that the authors bring up at page 7. Overall, what comes out is that there is this clean harmonic oscillation beating at 17 meV (in nice agreement with the electron-phonon coupling analysis of Giustino Nat. Comm. 7,11755), which actually points towards the lack of anharmonicities. Then, the power spectrum in Figure 3d, shows again broad features, pointing towards anharmonicities. So, is anharmonicity important or not? And why one gets cleaner signal from FAPbBr₃, which is supposed to be more disordered?

I personally suspect that there is just a mistake on the data treatment, as especially at 300 K one does not expect anything similar. The authors can compare with the power spectrum of the band gap, as a reference.

2) I think the authors should reduce the emphasis on the general importance of the A-site and may rather insist on the fact that single-photon emission is found at room temperature. Dealing with the A-site, the 72 meV redshift in the PL of FAPbBr₃ with respect to CsPbBr₃ is indeed marginal, especially when compared with the 700 meV blueshift with respect of CsPbI₃. However, I fully agree that this “marginal” effect has important implications in specific applications like single photon emission. In particular, an important outcome of the proposed localization mechanism is that it is effective at higher temperatures, where thermal motion is indeed more important. In this sense, the chemical message is partly “depleted” (what else can you use apart FA and MA?) but the mechanism itself of thermally induced charge localization may indeed guide the design of new materials, featuring this property.

3) I have some pain from time to time to connect with the procedures and results in Supporting Information: these are very well detailed but I must jump from the text to the SI to better understand what the authors do. I recommend the authors to slightly adjust this. For instance:

3.1) When mentioning the crystalline structure of materials (Figure 2a,b and Supporting Information), I recommend the authors to show the XRD patterns. I know the authors report a lot of data but sending the reader to the CCDC is very annoying.

3.2) I am not sure the authors mention anything about the statistics of the analysed AIMD snapshots. How many snapshots did you study? This will give a clear impression about how robust the statistic is (I know it is robust, as the videos and all the SI information suggest few hundreds snapshots).

3.3) When calling the results in Figure 3b and 3c, please add a reference to the employed procedure used to evaluate the size of the wavefunction and the autocorrelation function in the text (“see note #7” or “see Eq. S18”).

3.4) How do you get the power spectra in the inset of Figure 3c? just Fourier transforming the time-domain signal or is there anything else? Especially if you do something more complex, please, add a short note, either in the text or in Supporting Information,

Reviewer #3

(Remarks to the Author)

The work by Feld et al. reports high-purity ($g^{(2)}(0) < 0.05$), stable (> 1 hour), and bright (> 1 Mcps) single-photon emission from individual colloidal FAPbX₃ quantum dots with large spectral tunability. Through a combination of single-dot photoluminescence spectroscopy and ab initio molecular dynamics simulations, the authors attribute the suppressed biexciton emission to enhanced Auger recombination induced by charge-carrier localization. This study highlights the potential of organic-inorganic hybrid perovskite nanocrystals as high-performance quantum light sources operable under ambient conditions, and offers new fundamental insights into tailoring charge-carrier interactions in perovskite QDs beyond conventional size or shape control. I believe this work is of a quality suitable for publication in Nature Communications, pending clarification of several claims.

(1) In the section “Influence of the A-site cation on structure and optical properties” (Page 5), the authors state that CsPbBr₃ QDs and FAPbBr₃ QDs of almost the same size exhibit similar quantum confinement, despite their different exciton Bohr diameters (6.16 nm vs. 7.76 nm). However, the larger exciton Bohr diameter in FAPbBr₃ QDs would suggest stronger quantum confinement compared to CsPbBr₃ QDs, which in turn could lead to stronger charge-carrier localization. I would

like the authors to clarify whether this enhanced quantum confinement, in addition to that mediated by electron-phonon interactions, might also contribute to the more efficient biexciton Auger recombination and the higher single-photon purity observed here.

(2) In the same section (Page 5), the authors attribute the localized states to thermal disorder, which is somewhat unclear. The larger FA cation could better stabilize the crosslinked Pb–Br octahedra in their cubic phase, leading to less lattice distortion. Thermal disorder would therefore be more prominent in CsPbBr₃ QDs rather than in FAPbBr₃ QDs. I wonder if the disorder and the flatter Urbach tail could originate from excited-state lattice deformation and strong electron–phonon interactions in FAPbBr₃ QDs. Could the authors comment on this point?

(3) In the section “Phonon-driven wavefunction localization” (Page 6), the authors use 3.6 nm, ABr-terminated CsPbBr₃ and FAPbBr₃ QD models, both of which fall within the strong confinement regime, to analyze wavefunction localization in these two types of QDs. It would be helpful to clarify whether such models can accurately describe wavefunction localization in the experimentally studied QDs, which are in the weak confinement regime. It is worth noting that the wavefunction of HOMO in CsPbBr₃ extends notably to the QD surface even at 300 K (Fig. 3a and S8), indicative of strong quantum confinement. Clarification on how these models capture carrier localization under experimental conditions would strengthen the discussion.

(4) In the same section, the authors mainly analyze the spatial distribution of the HOMO in their QD models. However, since the experimental photoluminescence spectroscopy reflects the combined behavior of electrons and holes, it would be more appropriate to consider both the HOMO and LUMO levels simultaneously. Could the authors provide such an analysis?

(5) In the section “Purified single-photon emission” (Page 10), the authors attribute the quenched biexciton emission mainly to accelerated non-radiative (Auger) recombination in FAPbBr₃ QDs. However, according to a recent study (Angew. Chem. Int. Ed., 2020, 59, 14292–14295) based on power-dependent transient absorption spectroscopy measurements, CsPbBr₃ and FAPbBr₃ QDs of similar sizes exhibit comparable biexciton Auger recombination rates at room temperature, which are shorter than biexciton lifetimes determined by heralded single-particle photoluminescence spectroscopy. Could the authors clarify whether this discrepancy might arise from differences in measurement techniques?

(6) In the same section (Pages 10–11), the authors attribute the reduced photon anti-bunching at lower temperatures to enhanced wavefunction delocalization, which is said to significantly increase the radiative recombination rate of exciton species. However, biexciton Auger recombination could also be suppressed at low temperatures due to its thermally activated nature, as this non-radiative recombination requires energy and momentum conservation and is likely to be phonon-assisted. Therefore, the increased biexciton emissivity observed at low temperature might also arise from this effect. I recommend the authors clarify this point.

(7) In the section “FAPbX₃ QDs as single-photon sources” (Page 13), the authors demonstrate the wide spectral tunability of single-photon emission from FAPbX₃ QDs by controlling the halogen composition. However, phase segregation, as well as local environmental inhomogeneity even at the single-QD level, could introduce site-to-site variations in the emission properties of localized excitons, potentially leading to more significant spectral diffusion. Could the authors provide an outlook on possible technical approaches to suppress spectral diffusion in mixed-halide FAPbX₃ QDs?

Version 1:

Reviewer comments:

Reviewer #1

(Remarks to the Author)

Thank you for submitting the revised version of your manuscript. I appreciate the authors' careful and thoughtful responses to my previous comments. The revisions have improved the clarity and overall quality of the work.

Although a few of the issues I previously pointed out could not be fully resolved, the authors have provided reasonable explanations showing that these points fall outside the scope of the current study. I agree that addressing them in depth would exceed the intended focus of the manuscript and therefore does not affect its suitability for publication.

Overall, I am satisfied with the revisions and have no further concerns.

Reviewer #2

(Remarks to the Author)

The authors nicely addressed all my comments.

The manuscript, which was already excellent before the revision round where I was involved, further improved and reached a quality and clarity level worth of Nature Communication.

I congratulate with the author for the very good job

Reviewer #3

(Remarks to the Author)

I thank the authors for their clear and detailed responses to my comments. The revisions are thorough and significantly strengthen the manuscript. I believe the authors have addressed all concerns within the current technical limitations, and I recommend acceptance of the manuscript in its present form.

Response to the Referee Reports for Feld et al. (NCOMMS-25-40681-T)

We sincerely thank all the Reviewers for their detailed reading of the manuscript and their constructive comments. We followed their suggestions in close detail to improve the manuscript. Below, please find the revisions in response to the Reviewer's comments point-by-point, with the comments of the Reviewers in black, followed by our reply and action in blue.

We are convinced that our implemented changes have improved the manuscript and addressed the open points raised by the Reviewers.

Reviewer #1 (Remarks to the Author):

In this manuscript, the authors propose a novel strategy to enhance quantum confinement in colloidal quantum dots (QDs) and improve their single-photon purity, eliminating the need for QD downsizing. Based on combined theoretical and spectroscopic approaches, it is revealed that the A-site cation in soft APbBr₃ colloidal QDs controls the phonon-induced localization of the exciton wavefunction. Detailed density functional theory (DFT) results demonstrate that the transient wavefunction confinement is realized via strong exciton-phonon coupling, contrasting the generally negative impact of such couplings in optoelectronic materials. In my opinion, this work presents a solid piece of work, which should be of interest to the community. It sheds new light on achieving bright, stable, and pure single-photon emission, while circumventing the optical stability issues associated with conventional QD downsizing.

Our response: We are grateful for the appreciation and the attentive reading of our work.

I would recommend its publication pending the authors' consideration of the following points:

1. In the computational section, have the selected methods undergone benchmark work to verify their applicability to the target system? Additionally, how are the weak interactions accounted for?

Our response: Our computational methodology is based on recent work on phonon-induced dephasing and emission broadening in perovskite QDs (*Nat. Commun.* 2022, 13, 2587). We have opted for this method as it provides a well-proven balance between a sufficiently high level of theory and a sufficiently large model size, to best reflect the experimentally investigated QDs. This balance has previously facilitated to, often quantitatively, reproduce

several experimental observations made for perovskite QDs, from the emission linewidth to the carrier-cooling rates and the defect tolerance, see for example *ACS Photonics* 2020, 7, 5, 1088–1095, *Nat. Commun.* 2022, 13, 2587, and *ACS Energy Lett.* 2019, 4, 1, 63–74. In Figure R1, we summarize recent studies that showcase the high compatibility of this computational methodology and single-QD PL measurements. The agreement between experimental and simulated phonon frequencies as well as QD-size-dependent electron–phonon coupling demonstrates that this computational method is suitable for investigating electron–phonon coupling in soft perovskite compounds (*Adv. Optical Mater.* 2024, 12, 2301534; *Nat. Commun.* 2022, 13, 2587). In the present work, we additionally benchmarked our computational approach by comparing bandgap fluctuations in MD simulations and PL broadening in single-particle and ensemble PL studies (Supplementary Fig. 2-4). Also here, our computational models recover the experimentally observed temperature and A-site-cation dependence of phonon-induced energy fluctuations. Although still qualitative, given the small QD sizes computationally accessible, these past and present works demonstrate the suitability of the employed theoretical approach and its effectiveness in providing important insights into the photophysics governing exciton-phonon coupling.

As in the highlighted studies, we chose not to explicitly account for weak interactions in favour of larger computational models and longer MD trajectories. This choice is underpinned by reproducing various experimental trends at this level of theory and is additionally justified by the limited role of such interactions at room temperature. While such interactions affect geometry optimized (0 K) structures (*ACS Omega* 2020, 5, 40, 25723–25732) and are linked to the emergence of ordered nanodomains in otherwise disordered FAPbBr₃ and MAPbBr₃ crystals (*Nat. Nanotechnol.* 2025, 20, 755–763), such interactions cannot impose order over extended spatial or temporal ranges. This also becomes apparent in our X-ray total scattering experiments (Supplementary Fig. 20). Even at cryogenic temperatures, FAPbBr₃ lacks long range order (*J. Phys. Chem. Lett.* 2023, 14, 5, 1288–1293), suggesting a relatively weak interaction between the A-cation and Pb-Br frame. Further evidence comes from nuclear magnetic resonance of structurally-related cubic lead iodide perovskites, where the FA cations were observed to rotate invariantly on the picosecond-timescale (*J. Am. Chem. Soc.* 2023, 145, 2, 978–990).

Figure R1. Exciton-phonon coupling as seen by single-particle PL spectroscopy and by AIMD simulations. **a**, Representative single-particle PL spectrum from a sample of CsPbBr₃ QDs with 13 nm edge length at 4 K. **b**, Energy of the zero-phonon line (ZPL) and the energy of the phonon replica with respect to the ZPL, extracted from spectra as shown in **a**. **c**, Intensity of the phonon replica relative to the intensity of the ZPL, representing the magnitude of the exciton-phonon coupling strength. **d**, Reorganization energy as a function of the phonon energy obtained from AIMD simulations of a 1.8 nm CsPbBr₃ QD. **e**, Size-dependence of the coupling strength of the three phonon modes and the extrapolated room-temperature PL broadening. Note that the latter estimation neglects anharmonic effects that are relevant in this system at room temperature. Panels a-c were reproduced from *Adv. Optical Mater.* 2024, 12, 2301534; d,e were reproduced from *Nat. Commun.* 2022, 13, 2587.

2.A key highlight of this work is the achievement of high-purity single-photon emission without the need for QD downsizing. What criteria determine the optimal QD size for this mechanism? Is there potential for this mechanism to be applicable to bulk materials as well?

Our response: We appreciate the chance to address this point, because, there is indeed evidence that wavefunction localization exists in bulk and affects several photophysical properties (*J. Am. Chem. Soc.* 2022, 144, 41, 19137–19149, *Nano Lett.* 2015, 15, 1, 248–253).

However, wavefunction localization alone will not make bulk materials good single-photon emitters. While wavefunction localization relaxes momentum conservation rules, a first requirement for Meitner-Auger recombination, excitons also need to be coupled to each other (*Nano Lett.* 2010, 10, 1, 313–317). A sizable interaction strength, or biexciton binding energy, is required, which is not the case in bulk materials where excitons are independent. In other words, excitons created in the bulk do not collide and annihilate each other efficiently. Hence, bulk materials do not act as single-photon emitters. The stochastic nature of the absorption process generates a random number of excitons, which subsequently radiate a random number of photons, thereby precluding the antibunching feature characteristic of single-photon emitters.

Overall, we agree that there may be an optimal QD size for achieving stable and bright single-photon emission. Increasing the QD size enhances the brightness (via enhanced light absorption) and the photostability of the emitter. However, increasing the QD size eventually will also compromise the single-photon purity, since bulk systems are not effective two-level systems, as explained above. This suggests the existence of an optimal QD size for bright and stable QD-based single-photon emitters. While we refrain from providing here a precise number for the optimal QD size, our present work shows that good single-photon purity can, thanks to dynamic partial wavefunction localization, now be extended to QD sizes beyond 10 nm, *i.e.*, much larger than usual, yielding stable single-photon emission with a million counts per second (1 Mcps).

Our action: We clarified this point in the main text:

“Finally, temperature-induced localization even occurs in the respective bulk materials, specifically for FAPbBr₃ (Supplementary Fig. 11, 12). Although bulk crystals will not experience Auger-Meitner relaxation due to vanishing inter-exciton coupling, such localization effects reduce radiative and non-radiative rates and limit carrier mobilities in lead halide perovskites.^{45,58-60”}

3.As clarified in this manuscript, two types of QDs exhibit distinct phonon-driven wavefunction localization. Figure S3 in the Supporting Information presents the PL spectra of these two QDs as a function of temperature. I recommend that the authors conduct a quantitative analysis of the relationship between wavefunction localization and PL energy. Given that the PL energy shifts are generally regarded as directly related to wavefunction confinement, such a correlation could further validate the novel mechanism.

Our response: We appreciate this valuable suggestion. To follow it, we correlate the PL peak energy, PL peak width, and PL lifetime from Supplementary Fig. 3 with the wavefunction sizes from Figure 3b and Supplementary Fig. 5. As apparent in Figure R2, all three experimentally obtained metrics correlate with the wavefunction sizes, suggesting that these effects are

linked. PL peak width, PL lifetime and wavefunction localizations are directly linked to exciton-phonon coupling. We agree with your suggestion that the PL peak energy is sensitive to the wavefunction localization, and the analysis in Figure R2, indeed confirms our attribution that a stronger wavefunction localization is in place in FAPbBr₃ QDs (larger PL broadening, longer lifetime, larger PL bandgap shift). Nevertheless, there are several other temperature-sensitive influences that would need to be accounted for to fully describe PL peak energy temperature evolution. Those include optical and acoustic phonons (*Nanoscale* 2020,12, 13113-13118, *J. Phys. Chem. Lett.* 2021, 12, 1, 569–575), structural distortions (*Small Struct.* 2024, 5, 2300264, *J. Phys. Chem. Lett.* 2020, 11, 7, 2490–2496, *J. Phys. Chem. Lett.* 2021, 12, 1, 569–575), disorder (*npj Comput. Mater.* 2023, 9, 153), thermal expansion due to anharmonicity (*Phys. Chem. Chem. Phys.* 2020,22, 26069-26087), as well as phases transitions, for example occurring in FAPbBr₃ between 100 and 250 K (*Nano Lett.* 2018, 18, 7, 4440–4446, *Inorg. Chem.* 2018, 57, 2, 695–701, *J. Chem. Phys.* 2023, 158, 201104). The optical bandgap is, unfortunately, a complicated probe for temperature-induced wavefunction localization. A quantitative analysis is hindered by multiple effects, some of which are intertwined, making the attribution of wavefunction localization through this methodology possible only on a qualitative level.

Figure R2. Comparison of temperature-dependent PL characteristics and simulated wavefunction size. **a**, PL peak energy extracted from ensemble PL measurements. Grey arrows indicate phase transitions in FAPbBr₃ QDs (*Chem. Mater.* 2025, 37, 9, 3443–3454). **b**, PL peak width (full width at half maximum, FWHM) from ensemble PL measurements. **c**, PL lifetime (1/e decay time) from ensemble time-resolved PL. **d**, Average HOMO wavefunction size (given as FWHM) in MD simulations of 3.6 nm QD models.

4. It is found that the loss of correlation is significantly faster for the FAPbBr QD than for the CsPbBr₃ QD. Generally, the phonon-driven electronic dephasing will severely limit the prospects of QDs as single-photon sources. How should this issue be comprehensively addressed? While it does not undermine the novelty of the work.

Our response: We absolutely agree that such a temperature- and phonon-mediated mechanism is incompatible with applications that require long coherence time and

indistinguishability, such as photonic quantum computing. Nevertheless, there exist several quantum-enhanced communication or imaging schemes based on pure single photon sources which do not require such a property, e.g. QKD using the BB84 protocol. Such applications are not affected by dephasing and can be targeted by room-temperature single-photon sources (*Opt. Express* 2023, 31, 9437-9447, *Nat. Photonics* 2019, 116–122). If they offer sufficient photon count rates, single-photon purity and photostability for these applications, room-temperature single-photon sources may favour scalability and sustainability compared to cryogenically cooled single-photon sources. In this work, we propose and demonstrate a method that can enhance the single-photon purity in large, photostable QDs. We believe that marks an important step for colloidal QDs as room-temperature quantum light sources.

Reviewer #2 (Remarks to the Author):

The manuscript “Phonon-driven wavefunction localization promotes room temperature, pure single-photon emission in large hybrid lead halide perovskite quantum dots” by Feld and coworkers presents experimental evidences of the positive role from the dynamic disorder associated with the A-cation thermal motion on the single photon emission properties of halide perovskite quantum dots (QD). Namely, measurements of the second-order autocorrelation function of the photon arrival times ($g_2(t)$) at zero delay demonstrate single photon emission, at room temperature in CsPbBr₃ and FAPbBr₃ QDs, with single-photon purity of 71% and 96%, respectively. This is by itself an important results. The purity difference for the two compositions is then further rationalized using DFT-based Ab-Initio Molecular Dynamics (AIMD) simulations. The premise is that stronger confinement of the charge (as induced by reducing the QD size) improves single-photon purity (which I take as given, as I am not an expert myself in this specific field). From this premise, the authors provide convincing evidences from AIMD that the thermal motion in QD and related structural dynamic disorder localizes the electron/hole wavefunctions, ultimately acting as an additional confinement mechanism. Furthermore, they show that wavefunction localization is stronger in FAPbBr₃, as due to its specific dynamics. Thought I am not an expert in the field of single-photon emission, I enjoyed reading this manuscript, its shape well suiting the broad audience of the journal. The results are overall convincing and represent to my understanding a step forward in the current understanding of the problem. However, before recommending its publication on Nature Communication, I believe the authors should try to clarify an important point.

Our response: We really appreciate the positive assessment of our work.

As stated above, I like the message of the positive role of dynamic disorder on the single particle wavefunction confinement. Actually a similar mechanism was proposed long time ago by Wang (Nano Lett. 15, 248) and De Angelis (PCCP, 17,9394) to explain the reduced electron/hole recombination in 3D bulk halide perovskites. This however does not reduce the originality of the present work, as the charge localization is discussed here in relation to single photon-emission. However, the authors report indicative measurements of the wavefunction spread in Figure 3a,b, which set around 1-2 nm. On the other hand, when introducing the idea of spatial confinement from the exciton extension itself, in contrast to the QD size, the authors mention radii, the authors mention Bohr radii for the exciton of 6.16 nm and 7.76 nm, for CsPbBr₃ and FAPbBr₃, respectively. Even more, the Bohr radius is smaller for CsPbBr₃, while the results from AIMD point out stronger confinement from FAPbBr₃. This discrepancy somehow cast some shadow the full argument of the charge localization. Actually, it gives the impression that one compares “apples” with “pears”. Could the authors comment a bit on this? Why this discrepancy?

Our response: We appreciate the constructive feedback, and we acknowledge that introducing the Bohr diameter could be misleading. We mentioned the Bohr diameters to indicate that both samples would be in the intermediate confinement regime if one excluded the effect of disorder-induced wavefunction localization. Extracted from band structures, Bohr diameters indicate the exciton wavefunction size at 0 K. In weakly electronically disordered systems, the Bohr diameter can remain a good indicator of the exciton size or confinement also at finite temperature. In the presence of pronounced disorder at higher temperatures, however, this concept needs to be revised. The occupation of vibrational modes and their coupling to electronic degrees of freedom generate a disordered deformation potential that spatially and temporally fluctuates around a mean value (*Entropy* 2024, 26, 7, 552, *Phys. Rev. B* 2022, 106, 054311). This deformation potential acts as an additional effective confinement potential on the wavefunctions (*Phys. Rev. Lett.* 2016, 116, 056602). Therefore, at elevated temperature, FAPbBr₃ exhibits smaller wavefunctions than CsPbBr₃. We would like to point out that the reported wavefunction size extension is an instantaneous value extracted from DFT calculations. Given the dynamic nature of this localization, integration over many time realizations will broaden the spatial extent. However, it is this instantaneous localization that is responsible for the enhanced Auger rate and, consequently, the significantly improved single-photon emission.

Our action: We have clarified this point in the main text:

“Although the concept of Bohr diameters is not strictly valid at room temperature for materials exhibiting strong exciton-phonon coupling, we additionally corroborated the A-site-cation dependent single-photon purity for samples with identical sizes normalized by Bohr diameters (Supplementary Fig. 22).”

In addition, I take the occasion to rise few additional remarks, which are absolutely secondary but may still add something to the text.

1) I find the red signal in Figure 3 unexpectedly neat. For the sake of reference, the power spectrum of the band gap at 300 K of bulk halide perovskites was reported recently in *Adv. Opt. Mater.*, 2301105, 2023, showing very messy signals in the frequency domain. Notice also that such clean signal somehow contrasts a lot with the importance of phonon anharmonicity that the authors bring up at page 7. Overall, what comes out is that there is this clean harmonic oscillation beating at 17 meV (in nice agreement with the electron-phonon coupling analysis of Giustino *Nat. Comm.* 7,11755), which actually points towards the lack of anharmonicities. Then, the power spectrum in Figure 3d, shows again broad features, pointing towards anharmonicities. So, is anharmonicity important or not? And why one gets cleaner signal from FAPbBr₃, which is supposed to be more disordered? I personally suspect that there is just a mistake on the data treatment, as especially at 300 K one does not expect anything similar. The authors can compare with the power spectrum of the band gap, as a reference.

Our response: We appreciate this constructive comment on the data treatment and the effect of anharmonicity. We agree that the wavefunction autocorrelations and the associated power spectra are strikingly clean (they are obtained at 300 K) in comparison to bandgap power spectra in *Adv. Opt. Mater.* 2023, 2301105 or those reported by some of us (*Nat. Commun.* 2022, 13, 2587). By following the suggestion to construct such bandgap power spectra, we will show below that the displayed curves are not an artifact. However, we will first describe how the wavefunction autocorrelation and power spectra in Figure 3c were obtained. The absolute of the normalized molecular orbital coefficients $c_i(t)$ of the HOMO wavefunction recorded in NVE simulations (300 K, >14'000 steps, 1 fs timesteps) were first shifted by their average and then autocorrelated, followed by summation over all coefficients:

$$C_{\Phi}(\tau) = \int_{-\infty}^{\infty} dt \sum_i (|c_i(t)| - \langle c_i \rangle_t) (|c_i(t + \tau)| - \langle c_i \rangle_t). \quad (\text{eq. S9})$$

We multiplied this autocorrelation function with a \cos^2 -window function to minimize the accumulation of noise at long delay times: $C'_{\Phi}(\tau) = C_{\Phi}(\tau) \times \cos^2\left(\frac{\pi\tau}{L}\right)$, where L is the length of the MD trajectory. Finally, we normalized the autocorrelation function as $C_{\Phi,\text{norm}}(\tau) = C'_{\Phi}(\tau)/C'_{\Phi}(0)$, see Figure 3c. The power spectrum (Figure 3c, inset) was obtained as the absolute square of the Fourier transform of $C'_{\Phi}(\tau)$. In Figure R3a, we compare this analysis with and without applying a window function. This comparison convinced us that the window function does not cause the clean spectrum; it rather fulfills its purpose of enhancing the signal-to-noise ratio without significantly broadening spectral features.

Next, following your suggestion, we performed NVE simulations of our 3.6 nm QDs at 300 K and recorded the time-dependent bandgap (HOMO-LUMO gap). We recorded 2000 steps for both systems, employing 1 fs timesteps for FAPbBr₃ and 10 fs for CsPbBr₃. The corresponding autocorrelations and power spectra of the bandgaps are plotted in Figure R3b. We observe a high degree of similarity with the complex spectra shown in Adv. Opt. Mater. 2023, 2301105. We are therefore confident in the correctness of our procedure to obtain the power spectra of the bandgap and of the wavefunction coefficients. As for the difference between the power spectra of the bandgap and the wavefunction, we suspect that the multidimensionality of the wavefunction coefficients may enhance the signal-to-noise ratio.

Lead halide perovskites exhibit a high degree of anharmonicity, apparent for example in temperature-dependent crystal structures (Chem. Mater. 2025, 37, 9, 3443–3454) or short phonon lifetimes (Proc. Natl. Acad. Sci. U.S.A. 2018, 115, 47, 11905-11910). Pseudo-cubic perovskites are considered the most anharmonic amongst lead halide perovskites due to a shallow potential energy surface allowing for spontaneous local symmetry breaking (npj Comput. Mater. 2023,9, 153). The anharmonicity is also apparent by the broad low-frequency feature in the wavefunction power spectrum for FAPbBr₃, in stark contrast to CsPbBr₃, in which the less localized wavefunction is not influenced by an anharmonic mode, but by the Pb-Br stretching. Overall, the much broader power spectrum of the wavefunction autocorrelation in FAPbBr₃, alongside the significantly stronger disorder-enhanced single-photon purity and thermally increased PL lifetime in this material, suggest that anharmonicity is indeed an important ingredient for the disorder-induced wavefunction localization.

Our action: We have revised the description of the power spectrum analysis in the Supplementary Information (Supplementary Note 7.3), added Adv. Opt. Mater. 2023, 2301105 to our reference list, and clarified our discussion of anharmonicity in the computational section of the main text:

“The broad signal in FAPbBr₃, contrary to the narrow peak in CsPbBr₃, attests a high degree of anharmonicity.”

Figure R3. Comparison of wavefunction power spectra and bandgap power spectra at 300 K in 3.6 nm FAPbBr₃ and CsPbBr₃ QDs. a, Autocorrelation and power spectrum (inset) of the wavefunction coefficients and the effect of applying a cos²-window function, indicated by (w). **b,** Autocorrelation and power spectrum (inset) of the bandgap.

2) I think the authors should reduce the emphasis on the general importance of the A-site and may rather insist on the fact that single-photon emission is found at room temperature. Dealing with the A-site, the 72 meV redshift in the PL of FAPbBr₃ with respect to CsPbBr₃ is indeed marginal, especially when compared with the 700 meV blueshift with respect of CsPbI₃. However, I fully agree that this “marginal” effect has important implications in specific applications like single photon emission. In particular, an important outcome of the proposed localization mechanism is that it is effective at higher temperatures, where thermal motion is indeed more important. In this sense, the chemical message is partly “depleted” (what else can you use apart FA and MA?) but the mechanism itself of thermally induced charge localization may indeed guide the design of new materials, featuring this property.

Our response: We appreciate this valuable comment, and we agree that the control offered by this mechanism is an important result of this work with implications beyond lead halide perovskites. We would like to stress, however, that the marginal difference between the electronic structures of Cs- and FA-based QDs was indeed crucial for the present discovery and intended so. Although one may expect the two QD systems to exhibit similar electronic behavior (with Cs and FA orbitals not even participating in the electronic structure), their single-photon emission properties are markedly different, which we traced back to their differing crystal softness and exciton–phonon coupling strength. Having had at hand two electronically similar systems with markedly different structural dynamics and associated effects on the electronic wavefunction significantly strengthened our confidence in the main claim of this manuscript, i.e. the attribution of improved single-photon purity to stronger wavefunction localization. We highlight that all known organic (AZ, MA, FA)-inorganic

perovskite QDs possess better single photon purity at room temperature compared to fully inorganic perovskite QDs. As nicely outlined by you, we indeed hope that the results presented here can serve as the basis for new design rules in emerging material systems, opening pathways toward highly efficient quantum light sources operating at room temperature.

Our action: We have included your suggestion in the conclusions section of the main text:

“Considering the ubiquity of electron-phonon coupling, we are confident that actively tuning this interaction provides opportunities for functional material engineering beyond perovskite QDs.”

3) I have some pain from time to time to connect with the procedures and results in Supporting Information: these are very well detailed but I must jump from the text to the SI to better understand what the authors do. I recommend the authors to slightly adjust this. For instance:

Our response: We appreciate that these points are raised, and we have improved the readability and accessibility of our manuscript and data.

3.1) When mentioning the crystalline structure of materials (Figure 2a,b and Supporting Information), I recommend the authors to show the XRD patterns. I know the authors report a lot of data but sending the reader to the CCDC is very annoying.

Our response: The crystal structures were obtained from single crystal X-ray scattering data, which we have reported as normalized total scattering in Supplementary Fig. 20. The corresponding structure solutions are reported in Supplementary Table. 1-3. Such data is usually not represented as one-dimensional XRD patterns.

3.2) I am not sure the authors mention anything about the statistics of the analysed AIMD snapshots. How many snapshots did you study? This will give a clear impression about how robust the statistic is (I know it is robust, as the videos and all the SI information suggest few hundreds snapshots).

Our response: We appreciate raising this point. MD simulations in the NVT ensemble reported in Figure 3a,b employed timesteps of 1 fs. After an equilibration phase of 6 ps duration, we acquired trajectories for another 12-14 ps and printed cube files every 0.1 ps. Thus, over 120 snapshots were averaged per datapoint in Figure 3b. Similar analyses reported in the Supplementary Information employed 10 fs timesteps for CsPbBr₃ in some cases, e.g., bulk and 5.4 nm QDs. Simulations in the NVE ensemble reported in Figure 3c,d were performed after equilibration in the NVT ensemble. We employed 1 fs timesteps and acquired trajectories for > 14 ps while printing the HOMO wavefunction coefficients in each

MD frame. NVE-based trajectories reported in the Supplementary Fig. 4 employed 1 fs timesteps for FAPbBr₃ and 10 fs timesteps for CsPbBr₃.

Our action: We followed the suggestion to indicate the statistics in the main text and in methods section:

“Figure 3b shows the wavefunction sizes averaged across >120 snapshots (>12 ps; further details in Supplementary Note 7) in the CsPbBr₃ and FAPbBr₃ QDs along AIMD trajectories at 10, 100, and 300 K (further temperatures in Supplementary Fig. 5).”

“Trajectories with a duration of at least 12 ps were produced after equilibration at the respective temperature for 6 ps. To analyse the wavefunction confinement in real space, cube files for the highest occupied molecular orbital (HOMO) wavefunctions were extracted every 0.1 ps (>120 snapshots).”

3.3) When calling the results in Figure 3b and 3c, please add a reference to the employed procedure used to evaluate the size of the wavefunction and the autocorrelation function in the text (“see note #7” or “see Eq. SI8).

Our response and action: We followed this suggestion and included such references in the main text:

“Further insight into the phonon-driven wavefunction localization is provided by the autocorrelation function of the HOMO wavefunction coefficients which describes the time evolution of the wavefunction (Supplementary Note 7, Supplementary Equation 9).”

“The power spectra of the wavefunction autocorrelation (Supplementary Note 7) in the inset in Figure 3c reveal the dominant vibrational features driving the wavefunction localization in both systems.”

3.4) How do you get the power spectra in the inset of Figure 3c? just Fourier transforming the time-domain signal or is there anything else? Especially if you do something more complex, please, add a short note, either in the text or in Supporting Information,

Our response and action: We have provided a detailed description of the procedure in response to point 1) above and revised the description of the power spectrum analysis in the Supplementary Information (Supplementary Note 7.3)

Reviewer #3 (Remarks to the Author):

The work by Feld et al. reports high-purity ($g^{(2)}(0) < 0.05$), stable (> 1 hour), and bright (> 1Mcps) single-photon emission from individual colloidal FAPbX₃ quantum dots with large

spectral tunability. Through a combination of single-dot photoluminescence spectroscopy and ab initio molecular dynamics simulations, the authors attribute the suppressed biexciton emission to enhanced Auger recombination induced by charge-carrier localization. This study highlights the potential of organic–inorganic hybrid perovskite nanocrystals as high-performance quantum light sources operable under ambient conditions, and offers new fundamental insights into tailoring charge-carrier interactions in perovskite QDs beyond conventional size or shape control. I believe this work is of a quality suitable for publication in Nature Communications, pending clarification of several claims.

Our response: We are grateful for the positive evaluation of our work.

(1) In the section “Influence of the A-site cation on structure and optical properties” (Page 5), the authors state that CsPbBr₃ QDs and FAPbBr₃ QDs of almost the same size exhibit similar quantum confinement, despite their different exciton Bohr diameters (6.16 nm vs. 7.76 nm). However, the larger exciton Bohr diameter in FAPbBr₃ QDs would suggest stronger quantum confinement compared to CsPbBr₃ QDs, which in turn could lead to stronger charge-carrier localization. I would like the authors to clarify whether this enhanced quantum confinement, in addition to that mediated by electron-phonon interactions, might also contribute to the more efficient biexciton Auger recombination and the higher single-photon purity observed here.

Our response: We appreciate the opportunity to clarify this important point. In the absence of electron-phonon interaction, this small difference in size-confinement may lead to a small difference in single-photon purity. However, such a size difference is insufficient to explain the observed pronounced A-site-cation dependence of the single-photon purity: First, even at similar size-confinement d/d_B (dimensionless size; d : edge length, d_B : Bohr diameter), FAPbBr₃ QDs exhibit a higher single-photon purity (Figure R4). Secondly, there exists a strikingly different size dependence of the single-photon purity in these two materials (Figure 4c, Supplementary Fig. 21c). Both points demand an explanation that extends beyond the QD size, as we have provided with support of computational modelling, heralded single-particle PL spectroscopy, temperature-dependent single-particle PL spectroscopy, temperature-dependent time-resolved PL spectroscopy, transmission electron microscopy, and X-ray total scattering.

Our action: We have clarified this point in the main text and added Figure R4 as Supplementary Fig. 22 to the Supplementary Information:

“Although the concept of Bohr diameters is not strictly valid at room temperature for materials exhibiting strong exciton-phonon coupling, we additionally corroborated the A-site-cation dependent single-photon purity for samples with identical sizes normalized by Bohr diameters (Supplementary Fig. 22).”

Figure R4. Comparison of single-photon purity of FAPbBr₃ and CsPbBr₃ QDs (markers) and 95% confidence intervals (error bars) at identical confinement.

(2) In the same section (Page 5), the authors attribute the localized states to thermal disorder, which is somewhat unclear. The larger FA cation could better stabilize the crosslinked Pb–Br octahedra in their cubic phase, leading to less lattice distortion. Thermal disorder would therefore be more prominent in CsPbBr₃ QDs rather than in FAPbBr₃ QDs. I wonder if the disorder and the flatter Urbach tail could originate from excited-state lattice deformation and strong electron–phonon interactions in FAPbBr₃ QDs. Could the authors comment on this point?

Our response: We agree that H-bonding between the FA cation and the Pb-Br framework should have a stabilizing effect. In fact, a recent report linked such interactions to the emergence of ordered nanodomains in FAPbBr₃ and MAPbBr₃ (*Nat. Nanotechnol.* 2025, 20, 755–763). Yet, such interactions could not impose long range order, which also becomes apparent in our X-ray total scattering experiments (Supplementary Fig. 20). Even at cryogenic temperatures, FAPbBr₃ lacks long range order (*J. Phys. Chem. Lett.* 2023, 14, 5, 1288–1293), which suggests a relatively weak interaction between the A-cation and Pb-Br frame. Further evidence comes from nuclear magnetic resonance of structurally-related cubic lead iodide perovskites, where the FA cations were observed to rotate invariantly on the picosecond-timescale (*J. Am. Chem. Soc.* 2023, 145, 2, 978–990).

Spontaneous, dynamic, and local symmetry breaking through octahedral tilting, which is coupled with the A-site-cation reorientation in hybrid perovskites, appears as a global phenomenon in the high-temperature cubic phases of lead halide perovskites (*Nat. Nanotechnol.* 2025, 20, 755–763, *Joule* 2023, 7, 1051-1066, *ACS Energy Lett.* 2016, 1, 4, 880–887, *J. Phys. Chem. Lett.* 2017, 8, 19, 4720–4726, *J. Phys. Chem. C* 2023, 127, 38, 19141–

19151). Moreover, thermal disorder prevails over static disorder in these systems at room temperature (*J. Phys. Chem. Lett.* 2023, 14, 5, 1288–1293, *J. Phys. Chem. C* 2018, 122, 30, 17473–17480). With this body of literature and supported by our own experiments and simulations (Supplementary Fig. 15-18, 20), we are confident in our statement that FAPbBr₃ QDs exhibit more structural disorder at room temperature than CsPbBr₃ QDs.

That said, we share your opinion that the larger electronic disorder and more pronounced Urbach tails in FAPbBr₃ are also explained by a stronger exciton-phonon coupling. Electron-phonon coupling translates the structural disorder into electronic disorder. Various reports substantiate that FAPbBr₃ exhibits stronger electron-phonon coupling (*Nat. Phys.* 2024, 20, 47–53, *Nano Lett.* 2022, 22, 18, 7674–7681, *Adv. Optical Mater.* 2024, 12, 2301534). Consequently, electronic disorder and the associated wavefunction localization in FAPbBr₃ QDs are favoured by the thermal disorder as well as by electron-phonon coupling.

Finally, there is support for assigning the Urbach tails to phonon-induced wavefunction localization. Recent theoretical studies of disorder in structurally related MAPbI₃ found that the deformations by phonon-induced disorder are of similar magnitude as those caused by point defects (*J. Am. Chem. Soc.* 2022, 144, 41, 19137–19149). While polarons certainly play a key role in the photophysics of lead halide perovskites, the influence of thermal disorder on charge mobilities exceeds that of polarons (*Phys. Rev. Lett.* 2020, 124, 196601) to which reduced charge mobilities were often attributed (*Sci. Adv.* 2017, 3, e1701217, *Nat. Commun.* 2016, 7, 12253).

Our action: We clarified the discussion of the Urbach tail:

“Moreover, we observed an exponential low-energy (Urbach) tail in single-particle PL spectra, see Figure 2f. Across a large size range, the tails are consistently steeper for CsPbBr₃ than in comparably sized FAPbBr₃ QDs (Figure 2g). As elucidated in a prior study,⁴⁸ this suggests stronger exciton-phonon coupling in FAPbBr₃ QDs, consistent with PL phonon replica at cryogenic temperatures^{53,54} and optical-pump–electron-diffraction-probe measurements.⁵⁵ This trend is also confirmed by our accompanying density functional theory (Supplementary Fig. 4) and ensemble PL studies (Supplementary Fig. 3). Urbach tails are associated with exciton-phonon coupling through the formation of localized states that result from thermal disorder and form the low-energy tail.^{32,34,36} They thus also serve as a first indication of the hypothesized wavefunction confinement which may be enhanced in pseudo-cubic FAPbBr₃ QDs.”

(3) In the section “Phonon-driven wavefunction localization” (Page 6), the authors use 3.6 nm, ABr-terminated CsPbBr₃ and FAPbBr₃ QD models, both of which fall within the strong confinement regime, to analyze wavefunction localization in these two types of QDs. It would be helpful to clarify whether such models can accurately describe wavefunction localization

in the experimentally studied QDs, which are in the weak confinement regime. It is worth noting that the wavefunction of HOMO in CsPbBr₃ extends notably to the QD surface even at 300 K (Fig. 3a and S8), indicative of strong quantum confinement. Clarification on how these models capture carrier localization under experimental conditions would strengthen the discussion.

Our response: We appreciate your comment about the distinct size ranges covered by QD models and by experiments. While indeed unfortunate, such a remaining size disparity is the state of the art in the field when comparing single-QD measurements and ab-initio methods. On the one hand, experimentally measuring single QDs becomes increasingly more challenging for decreasing QD size, especially for hybrid organic-inorganic variants, and is typically only performed for perovskite QDs larger than about 4 nm. Moreover, ab-initio methods, as required here to quantify wavefunction localization, intrinsically come with a very high computational cost. For example, for each computational data point in Figure 3b, we have committed nearly 10⁶ CPU hours to obtain ~20 ps long trajectories of atomistic FAPbBr₃ QD models in the canonical ensemble. The same effort was put towards the NVE simulations for Figure 3c. The high cost originates from the large system size (3652 atoms) and the small timesteps (1 fs) required to capture fast intramolecular vibrations of the organic A-cation. With currently available computational methods, there is no shortcut circumventing this bottleneck, as our interest in the global wavefunctions of the QD precludes cost-cutting methods such as linear-scaling DFT which imposes a fragmentation of the system (*J. Phys. Chem. A* 2023, 127, 3, 589–618).

Despite the partially incomplete match between computationally and experimentally assessed QD size ranges, several recent studies successfully combined computational QD modelling and single-QD PL spectroscopy, particularly to assess exciton-phonon coupling. For instance, there is great agreement in electron-phonon coupling spectra between single-QD PL measurements at cryogenic temperatures and computational modelling, delivering matching phonon mode frequencies and QD size dependencies (Figure R5; *Adv. Optical Mater.* 2024, 12, 2301534; *Nat. Commun.* 2022, 13, 2587). In the present work, we further benchmarked our computational approach by comparing bandgap fluctuations in MD simulations and PL broadening in single-particle and ensemble PL studies (Figures S2, S3, and S4). We found that our computational models recover both the temperature and A-cation dependence of phonon-induced energy variations.

To account for the QD-size dependence within the given computational-cost constraints, our original submission quantified the extent and dynamics of the wavefunction for QD sizes up to 3.6 nm for FAPbBr₃ and up to 5.4 nm for CsPbBr₃ (Supplementary Fig. 6, 8). We found temperature-induced localization across all QD sizes. In previous studies (*Nature* 2024, 626, 535–541, *Nat. Commun.* 2022, 13, 2587, *Nano Lett.* 2020, 20, 3, 1819–1829), such QD size

dependencies proved instrumental in establishing a qualitative agreement between AIMD simulations and experimental observations. Here, we find that the wavefunction size in CsPbBr₃ QDs depends on the QD size while that of FAPbBr₃ QDs does not. This observation supports your suggestion of size confinement in CsPbBr₃, while it suggests a disorder-limited wavefunction size in FAPbBr₃. It also aligns with the experimentally observed weakly size-dependent single-photon purity in FAPbBr₃ and a stronger size dependence in CsPbBr₃.

Finally, we have eliminated any potential overestimation of finite size effects by monitoring the wavefunctions in simulations of bulk crystals, employing periodic boundary conditions. We observe similar temperature and A-site-cation dependencies of the wavefunction localization (Supplementary Fig. 9). Such bulk simulations of wavefunction localization have previously also explained long radiative and non-radiative charge carrier lifetimes (*J. Am. Chem. Soc.* 2022, 144, 41, 19137–19149, *Nano Lett.* 2015, 15, 1, 248–253).

Taken together, we have benchmarked our computational-experimental approach for the study of exciton-phonon coupling in perovskite QDs before its application to wavefunction localization. Additionally, there exist numerous studies that have successfully taken a similar route to decode exciton-phonon coupling in general and wavefunction localization in particular, demonstrating a solid link to experimental observations.

Our action: We have now addressed this point in the main text of the revised manuscript:

“Such a systematically observed wavefunction localization across model sizes ranging from 1.8 nm QDs to the bulk alongside the qualitative agreement in key experimental observations linked to exciton-phonon coupling (Supplementary Fig. 3, 4),^{7,26} affirms that such wavefunction localization also occurs in the perovskite QD sizes accessed experimentally (vide infra).”

Figure R5. Exciton-phonon coupling as seen by single-particle PL spectroscopy and by AIMD simulations. **a**, Representative single-particle PL spectrum from a sample of CsPbBr₃ QDs with 13 nm edge length at 4 K. **b**, Energy of the zero-phonon line (ZPL) and the energy of the phonon replica with respect to the ZPL, extracted from spectra as shown in **a**. **c**, Intensity of the phonon replica relative to the intensity of the ZPL, representing the magnitude of the exciton-phonon coupling strength. **d**, Reorganization energy as a function of the phonon energy obtained from AIMD simulations of a 1.8 nm CsPbBr₃ QD. **e**, Size-dependence of the coupling strength of the three phonon modes and the extrapolated room-temperature PL broadening. Note that the latter estimation neglects anharmonic effects that are relevant in this system at room temperature. Panels a-c were reproduced from *Adv. Optical Mater.* 2024, 12, 2301534; d,e were reproduced from *Nat. Commun.* 2022, 13, 2587.

(4) In the same section, the authors mainly analyze the spatial distribution of the HOMO in their QD models. However, since the experimental photoluminescence spectroscopy reflects the combined behavior of electrons and holes, it would be more appropriate to consider both the HOMO and LUMO levels simultaneously. Could the authors provide such an analysis?

Our response: We recognize that including the LUMO (electron wavefunction) in our analysis is crucial and we accept the challenge to do so. We identified two approaches to model the

notoriously difficult LUMO wavefunction and demonstrate that the LUMO, similarly to the HOMO, indeed also experiences a thermally activated wavefunction localization, hereby supporting the proposed link between thermal disorder and single-photon purity. We appreciate the constructive comment which strengthened the conclusions of our work by clarifying the role of the electron wavefunction.

The key to analyse the LUMO wavefunction lies in passivating the perovskite QD surface. Perovskite QD models without full surface passivation create surface-localized LUMOs due to undercoordinated bromine atoms, see Figure R6a as well as previous works on the topic, e.g. *ACS Energy Lett.* 2019, 4, 1, 63–74 and *J. Am. Chem. Soc.* 2022, 144, 25, 11059–11063. While the atomistic understanding of the surface termination has recently experienced help from nuclear magnetic resonance spectroscopy (*J. Am. Chem. Soc.* 2020, 142, 13, 6117–6127, *Nature* 2024, 626, 542–548), the exact ligand coverage and ligand arrangement remains elusive. This insufficient understanding of the precise surface chemistry hinders us from passivating the QD surface with organic surface-capping ligands in exact analogy to the experiment. Currently, efforts are underway to introduce a chemically and physically satisfying passivation in our models and experimental validation is ongoing.

For the time being, the electrostatic environment provided by surface ligands can be emulated by epitaxially shelling CsPbBr₃ with a CsCaBr₃-shell, as shown in Figure R6b. This wide-bandgap shell enables the delocalization of the LUMO wavefunction in the QD core, as we could show in two recent studies (*Nat. Commun.* 2022, 13, 2587, *Nature* 2024, 626, 535–541). Such an approach also successfully reproduces the temperature-dependent wavefunction (de)localization observed by single-particle photoluminescence experiments in conjunction with k.p modelling (Extended Data Figure 4 and Supplementary Video 2 in *Nature* 2024, 626, 535–541). Here, we employ this surface passivation for CsPbBr₃ to provide a full analysis of the LUMO wavefunction size, summarized in Figure R6. As in the assessment of the HOMO wavefunction, we equilibrated the QD models at the target temperature for 6 picoseconds and performed MD simulations in the canonical ensemble for another 14 picoseconds. Figure R6c displays representative snapshots of the LUMO wavefunction at 100 K and 300 K and Figure R6b reports the mean wavefunction sizes. We also included a video of the LUMO density as Supplementary Video 8. It becomes clear that the LUMO wavefunction, like the HOMO, contracts as thermally induced dynamic disorder emerges. Furthermore, being in CsPbBr₃ and not FAPbBr₃, the effect remains relatively small. The relative change in wavefunction size correlates well with that of the HOMO wavefunction in a bare 2.4 nm CsPbBr₃ QD (Supplementary Fig. 7) suggesting a roughly comparable vibronic coupling for electron and hole. While this data so far fully aligns with our theory, a comparison between CsPbBr₃ and FAPbBr₃ is not possible. A shelling material that is dielectric and lattice-matched, like CsCaBr₃ for CsPbBr₃, is yet to be identified for FAPbBr₃.

To offer such a comparison of CsPbBr₃ and FAPbBr₃, we turn to simulations under periodic boundary conditions, mimicking the respective bulk materials. We reported in our original submission that the disorder-induced wavefunction localization also occurs under periodic boundary conditions. In line with QD models, the HOMO wavefunction size was temperature and A-cation dependent, experiencing stronger localization in FAPbBr₃ than in CsPbBr₃ (Supplementary Fig. 12). Eliminating any surface effects, these bulk studies provide the opportunity to fully address your concern and assess the LUMO wavefunction localization in both FAPbBr₃ and CsPbBr₃. Our main results of the analysis are summarized in Figure R7, while videos of the LUMO density are provided as Supplementary Videos 6 and 7. Representative snapshots of the LUMO wavefunctions in FAPbBr₃ and CsPbBr₃ at 100 K and 300 K show that the wavefunctions localize upon temperature-increase and the localization is stronger in FAPbBr₃ than in CsPbBr₃ (Figure R7a). This initial observation is confirmed by quantifying the wavefunction localization as the inverse participation ratio (Figure R7b). We therefore find that the temperature and the A-cation dependence of the disorder-induced wavefunction localization holds also for the LUMO wavefunction.

Our action: We have included Figure R6 and R7 as Supplementary Fig. 10 and 11 in the updated Supplementary Information alongside Supplementary Note 8. Videos of LUMO wavefunctions are provided as Supplementary Videos 6-8. The LUMO wavefunction localization is cited in the main text:

“The electron wavefunction, represented by the lowest unoccupied molecular orbital (LUMO), experiences similar wavefunction localization as the HOMO (Supplementary Note 8, Supplementary Fig. 10, 11). Finally, temperature-induced localization even occurs in the respective bulk materials, specifically for FAPbBr₃ (Supplementary Fig. 11, 12).”

Figure R6. Temperature-induced LUMO wavefunction localization in AIMD simulations of a 3.6 nm CsPbBr₃/CsCaBr₃ core/shell QD. **a**, Representative LUMO wavefunction in a 3.6 nm CsPbBr₃ QD at 100 K. **b**, Cross-section of a 3.6 nm CsPbBr₃/CsCaBr₃ core/shell QD with a 2.4 nm CsPbBr₃ core. **c**, Representative LUMO wavefunctions of a 3.6 nm CsPbBr₃/CsCaBr₃ core/shell QD at 100 and 300 K. **d**, Average LUMO wavefunction sizes (full width at half maximum, markers) and 95% confidence intervals (error bars) at 100 K and 300 K.

Figure R7. Temperature-induced LUMO wavefunction localization in AIMD simulations of bulk FAPbBr₃ and CsPbBr₃. **a**, Representative LUMO wavefunctions at 100 and 300 K. **b**, Mean inverse participation ratio (IPR) as a function of temperature. Error bars indicate 95% confidence intervals.

(5) In the section “Purified single-photon emission” (Page 10), the authors attribute the quenched biexciton emission mainly to accelerated non-radiative (Auger) recombination in FAPbBr₃ QDs. However, according to a recent study (Angew. Chem. Int. Ed., 2020, 59, 14292–14295) based on power-dependent transient absorption spectroscopy measurements, CsPbBr₃ and FAPbBr₃ QDs of similar sizes exhibit comparable biexciton Auger recombination rates at room temperature, which are shorter than biexciton lifetimes determined by heralded single-particle photoluminescence spectroscopy. Could the authors clarify whether this discrepancy might arise from differences in measurement techniques?

Our response: We thank you for this important point. While it is difficult to comment on other works, we do share your opinion that such apparent discrepancy could arise from the different technique or averaging effect at the ensemble level.

We have revealed faster biexciton decay in FAPbBr₃ QDs from a direct measurement of the biexciton lifetime in both systems via heralded single-QD PL spectroscopy. Heralded spectroscopy, by the fact that it post-selects for emitted pairs, naturally excludes contributions from dim emitting states and from charged states which may affect ensemble transient absorption measurements and would appear as shorter Auger decay times.

(6) In the same section (Pages 10–11), the authors attribute the reduced photon anti-bunching at lower temperatures to enhanced wavefunction delocalization, which is said to significantly increase the radiative recombination rate of exciton species. However, biexciton Auger recombination could also be suppressed at low temperatures due to its thermally activated nature, as this non-radiative recombination requires energy and momentum conservation and is likely to be phonon-assisted. Therefore, the increased biexciton emissivity observed at low temperature might also arise from this effect. I recommend the authors clarify this point.

Our response: We appreciate this comment. We agree that the temperature dependence of the biexciton quantum yield could be affected also by a thermally activated nature of Auger-Meitner recombination (*J. Phys. Chem. C* 2010, 114, 41, 17550–17556, *J. Phys. Chem. Lett.* 2014, 5, 1, 99–105), although we want to point out that a consensus on the universal presence of such effects has not been reached, see e.g. *J. Phys. Chem. Lett.* 2020, 11, 16, 6513–6518. Disentangling the relative contributions to the temperature-dependent single-photon purity is challenging, as these effects exhibit a similar temperature dependence. However, acquired at the same temperature, the A-site-cation dependent experiments substantiate our conclusions.

Our action: We have refined the discussion of the temperature dependence to account for the combined effects of thermally activated Auger recombination and disorder-induced wavefunction localization:

“Matching the temperature dependence of the wavefunction localization (Figure 3b), the single-photon purity increases with increasing temperature in both materials (Supplementary Fig. 21, 24). It is noteworthy that this temperature dependence could in parts be linked to single-photon superradiance, a process associated with a wavefunction delocalization that causes a rapid radiative decay and that is lost at higher temperatures due to thermal disorder,⁷ or to Auger-Meitner recombination intrinsically requiring thermal activation.^{73,74} Nevertheless, a stronger increase of single-photon purity observed in FAPbBr₃ demonstrates that disorder-induced wavefunction localization plays a key role (Supplementary Fig. 21, 22, 24). These temperature and composition trends are also followed by the increasing PL peak width, an indicator for exciton-phonon coupling, and the increasing PL lifetime

(Supplementary Fig. 3), which was recently linked with wavefunctions localized through vibrations.^{58,59} All these effects reflect the disorder-induced wavefunction localization.”

(7) In the section “FAPbX3 QDs as single-photon sources” (Page 13), the authors demonstrate the wide spectral tunability of single-photon emission from FAPbX3 QDs by controlling the halogen composition. However, phase segregation, as well as local environmental inhomogeneity even at the single-QD level, could introduce site-to-site variations in the emission properties of localized excitons, potentially leading to more significant spectral diffusion. Could the authors provide an outlook on possible technical approaches to suppress spectral diffusion in mixed-halide FAPbX3 QDs?

Our response: We acknowledge that the wavefunction localization, subject of this work, in liaison with other forms of heterogeneities can give rise to altered optical characteristics. The mentioned heterogeneity in mixed halide perovskites could lead to particularly complex and fascinating features. To the best of our knowledge, there currently exists no consensus regarding the halide distribution in perovskite QDs in equilibrium or under irradiation. Moreover, we could not identify conclusive proof of phase segregation as observed in the bulk. We would thus need to speculate on the interplay of phonon-induced wavefunction localization and heterogeneity in halide distribution. While a deeper analysis would be required to provide a definitive conclusion, we could so far not spot significantly enhanced spectral diffusion in our data. It is noteworthy that spectral diffusion is largely masked by the homogeneous linewidth, which dominates PL peak broadening of perovskite QDs in single-particle PL measurements at room-temperature (*Nat. Commun.* 2022, 13, 2587).